materials science

hemp fibre, quaternary ammonium groups, covalent grafting, antibacterial activity

**Authors for correspondence:**
Lining Zhao
e-mail: csbtzln@163.com
Defang Li
e-mail: 13873129468@126.com

This article has been edited by the Royal Society of Chemistry, including the commissioning, peer review process and editorial aspects up to the point of acceptance.

# Improved antibacterial activity of hemp fibre by covalent grafting of quaternary ammonium groups

Li Chang[1], Wenjie Duan[2,3], Siqi Huang[1], Anguo Chen[1], Jianjun Li[1], Huijuan Tang[1], Gen Pan[1], Yong Deng[1], Lining Zhao[1] and Defang Li[1]

[1]Institute of Bast Fibre Crops, Chinese Academy of Agricultural Sciences, 410205 Changsha, Hunan, People's Republic of China
[2]Institute of Chemistry, Henan Academy of Sciences, 450003 Zhengzhou, Henan, People's Republic of China
[3]School of Materials Science and Engineering, Zhengzhou University, 450000 Zhengzhou, Henan, People's Republic of China

LC, 0000-0002-9513-7637; LZ, 0000-0001-8182-4818; DL, 0000-0002-1022-4046

In this study, a novel antibacterial hemp fibre grafted with quaternary ammonium groups (HF–GTA), were prepared by alkalization, oxidation, amination and quaternization multistage reactions. The chemical structure and micromorphology of the fibre were characterized by Fourier-transform infrared spectroscopy, scanning electron microscopy, X-ray photoelectron spectroscopy, thermogravimetric analysis and X-ray diffraction. The grafting and reaction mechanism proved to be successful, which indicated that the grafting reaction primarily occurred on the hydroxyl group of cellulose and hemicellulose in the hemp fibre, where it retained good fibrous morphology, thermal stability and hygroscopicity. HF–GTA exhibited the best antibacterial activity, where the antibacterial ratios against *Escherichia coli* and *Staphylococcus aureus* were 95.41% and 99.64%, respectively. Even after washing 30 times, the antibacterial activity was retained at 89.78% and 91.12%, indicating that HF–GTA was endowed with good washing resistance. The antibacterial activity was owing to the electrostatic reaction reducing the electrochemical potential on the cell membrane, leading to the release of cytoplasmic substances and the dissolution of cells. This study is significantly important for guaranteeing textile quality and preventing disease transmission.

# 1. Introduction

Harmful microorganisms are prolific in daily life. These microorganisms cannot only cause disease and death in humans and animals but also cause the decomposition and deterioration of various materials. Textiles are excellent habitats for these microorganisms and an important source of disease transmission. The production of textiles with antimicrobial properties is increasingly common and has become an area of significant research focus. Finishing is a traditional method; however, the resultant textiles have poor antimicrobial durability. With the development of science and technology, researchers have begun to use modification methods to produce fibres with long-lasting antimicrobial effects. In modification methods, antibacterial agents are introduced into the surface and the inside of the fibres by physical or chemical methods [1–6]. Because of the strong binding between the antibacterial agents and fibres, the resulting antibacterial fibres maintain long-lasting antibacterial effects.

Hemp (*Cannabis sativa* L.) is an annual herb, which has been dispersed and cultivated by humans across the world, from the tropics to alpine foothills. It is one of the oldest plant sources for seed oil, intoxicant resin, medicine and even textile fibres, trailing back at least six millennia [7]. For a few millennia, hemp has been a respected crop in China, where it became an important fibre for clothing [8,9]. Because hemp has a low requirement for fertilizers and pesticides during planting, the harvested fibre can be used as an excellent environmental textile fibre with rich resources. More importantly, compared with other fibres, hemp fibre has specific properties, such as aseptic properties, high absorbency and hygroscopicity, good thermal and electrostatic properties, protection against UV radiation, lack of any allergenic effect, and low cost [10–12]. Furthermore, hemp fibre is very soft and has a blunt end that prevents sharp itching. Nowadays, clothing made of hemp fibre is gradually becoming favourable, and its market demand is accelerating [9]. According to recent studies, hemp bast fibre demonstrated some antibacterial properties because of its hollow porous morphology and antibacterial substances that can destroy the living environment of anaerobic bacteria [13]. Nevertheless, hemp bast fibre contains a large amount of gum. Therefore, it is necessary to degum hemp bast fibre for textiles [14]. This process causes the loss of most of the antibacterial components, resulting in the loss of the antibacterial property of hemp fibre; therefore, it is difficult to meet the human demand for antibacterial hemp fibre.

The structure of hemp fibre comprises three major constituents: cellulose, hemicellulose and lignin. Cellulose is considered to be the major framework component of the fibre structure. The chemical structure of cellulose contains three hydroxyl groups which form hydrogen bonds in the macromolecular cellulose network. Two of these hydroxyl groups form inter-molecular bonds, while the third forms intra-molecular hydrogen bonds [15]. These hydroxyl groups can be used as reactive groups for a series of chemical reactions, such as oxidation, esterification, etherification, swelling and grafting reactions [16,17]. Therefore, hemp fibre is an ideal raw material as an easily renewable source, with the potential for modification to prepare functional antibacterial fibres. It is significative to endow hemp fibre with antibacterial properties to produce an important functional material that can guarantee textile quality and prevent disease transmission.

Quaternary ammonium salts are widely used as efficient disinfectants in medical and healthcare, food, daily chemicals, and other fields, and are currently the most studied organic antibacterial agents. Quaternary ammonium salts contain cations, while bacteria usually contain anions on their surface. Therefore, it is easily adsorbed on the fibre surface. However, the antibacterial agents of quaternary ammonium salts are only adsorbed on the surface of cellulose; hence, they are easily lost during use, thereby causing environmental pollution [18]. Therefore, quaternary ammonium functional groups have been covalently grafted onto the surface of cellulose, thereby greatly improving the antibacterial properties and stability. For example, Liu *et al.* [19] grafted three kinds of quaternary ammonium salts onto a chitosan nanocomposite membrane via etherification that displayed excellent biocidal abilities against both *Staphylococcus aureus* and *Escherichia coli*. Oyervides-Muñoz *et al.* [20] synthesized three different ammonium salts with a carboxylic acid end group through a quaternization reaction and chemically grafted them along the chitosan backbone using 1-ethyl-3-(3-dimethylaminopropyl) carbodiimide as a carboxyl-activating agent for coupling with the primary amine groups, which significantly improved the antibacterial activity against *Pseudomonas aeruginosa*. Xu *et al.* [21] prepared a transparent cellulose membrane grafted onto the different ratios of sulfobetaine methacrylate and [2-(Acryloyloxy)ethyl]trimethylammoniumchlorid via surface-initiated atom-transfer radical polymerization. Owing to the presence of quaternary ammonium groups, the antibacterial properties of the cellulose membrane were improved, where the best antibacterial ratios were 95.1% against *S. aureus* and 90.5% against *E. coli*.

To date, studies based on the modification of hemp fibre for antibacterial application have been rarely reported. In this study, we report the novel preparation of functional, antibacterial hemp fibre by the

covalent grafting of quaternary ammonium groups. The as-prepared hemp fibre modified by quaternary ammonium groups (HF–GTA) presented favourable antibacterial activities against Gram-positive and Gram-negative bacteria. The properties of the hemp fibre before and after modification was characterized by Fourier-transform infrared (FTIR) spectroscopy, X-ray photoelectron spectroscopy (XPS), scanning electron microscopy (SEM), X-ray diffraction (XRD) and thermogravimetric analysis (TGA). The obtained HF–GTA remained advantageously fibrous, which results in more resourceful models in practice. For example, it could be used in filaments or staples or woven in thread and cloth.

# 2. Material and methods

## 2.1. Materials and reagents

Degummed hemp fibre (cellulose > 90%, hemicellulose 2–3%, lignin < 1%, pectin < 1%, lipids < 1%) with a length of approximately 5 cm and the metric number of 1568 Nm was obtained from Hunan Huasheng Group Co., Ltd., China. Sodium periodate ($NaIO_4$), triethylenetetramine (TETA), glycidyl trimethyl ammonium chloride (GTA), 50% glutaraldehyde and hydroxylamine hydrochloride were purchased from Shanghai Macklin Biochemical Technology Co., Ltd., China. Sodium chloride, sodium hydroxide, glycerol, methanol, ethanol, ethylene glycol, propylene glycol and acetone were obtained from Hunan Huihong Reagent Co., Ltd., China. Thymol blue were purchased from Tianjin Kermel Chemical Reagent Co., Ltd., China. Peptone and beef extracts were purchased from Beijing Solarbio Science & Technology Co., Ltd., China. Agar was obtained from BioFROXX Biotechnology Co., Ltd., China. All chemicals and reagents used were of analytical grade and used as received. Pathogenic *E. coli* and *S. aureus* were supplied by the Institute of Biology, Henan Academy of Sciences, Co., Ltd., Zhengzhou, China. Distilled water was used for the experiments.

## 2.2. Preparation of hemp fibre grafted with quaternary ammonium groups

Four steps were used to prepare the antibacterial hemp fibre (HF–GTA). The first step was an alkalization reaction. The hemp fibre, used as the parent fibre, was soaked in 1–30% NaOH solution for 10–180 min at room temperature (20–30°C), and the fibre was washed with distilled water until the pH became neutral. The obtained alkaline fibre HF–OH was dried at 40°C to a constant weight. Secondly, an oxidation reaction was performed. The HF–OH was reacted with 0.5–3% $NaIO_4$ solution at 30–70°C in the dark at pH 3.5–11 for 20–180 min, and then the fibre was immersed in 1% glycerol solution for 30 min to remove the unreacted $NaIO_4$. The obtained oxidized fibre HF–CHO was fully washed with distilled water and dried at 40°C to constant weight. Thirdly, an amination reaction was performed. The HF–CHO was reacted with 0.25 ml TETA at 30–50°C in glycerol, propylene glycol, ethylene glycol, ethanol, acetone, distilled water and pure TETA for 1–12 h, during the reaction, 500 µl of 50% glutaraldehyde was added dropwise within 1 h, and then the aminated fibre $HF–NH_2$ was washed with distilled water and dried at 40°C to constant weight. Finally, the quaternization reaction was performed. The $HF–NH_2$ was reacted with 0.2–10% GTA at 30–70°C in acetone, distilled water, glycerol, propylene glycol, ethylene glycol and ethyl alcohol for 5–60 min, and then the final quaternized fibre HF–GTA was washed with distilled water and dried at 40°C to constant weight. The weight gain was determined by equation (2.1). The reaction mechanism is shown in figure 1.

$$\text{Weight gain (\%)} = \frac{W_f - W_0}{W_0} \times 100, \tag{2.1}$$

where $W_0$ is the mass of fibre before modification and $W_f$ is the mass of fibre after modification. It should be noted that when the calculated weight gain (%) is negative, the absolute value is recorded as the weight loss (%).

The content of the aldehyde group was determined using the method of hydroxylamine hydrochloride [22,23] with minor modifications as follows: 3–5 drops of thymol blue indicator were added to 50 ml of 20 g l$^{-1}$ hydroxylamine hydrochloride methanol solution. If the solution turned red, a sodium hydroxide methanol solution of 0.03 mol l$^{-1}$ was added dropwise to make it yellow; then the fibre HF–CHO (0.1500 g) was added into the above solution and oscillated at 180 r.p.m. for 30 min at room temperature (20–30°C). The fibre was removed, and the solution was titrated with 0.03 mol l$^{-1}$ sodium hydroxide methanol solution to yellow.

$$W = \frac{30 \times V}{m}, \tag{2.2}$$

**Figure 1.** Reaction mechanism for the synthesis of antibacterial hemp fibre.

where $W$ is the content of the aldehyde group (mol g$^{-1}$), $V$ is the volume of sodium hydroxide methanol solution consumed (ml), and $m$ is the mass of oxidized fibre (g).

## 2.3. Characterization

FTIR spectra were recorded with a Bruker Invenio spectrometer (Invenio, Germany) with a Platinum ATR module (equipped with a diamond crystal), with wavenumber ranging from 400 to 4000 cm$^{-1}$ and resolution of 4 cm$^{-1}$; and each sample was scanned 16 times. XPS spectra were analysed by an Escalab 250XiXPS with a monochromatic Al–Kα X-ray source. SEM (JSM-7500F) was used to observe the surface morphology of the fibres before and after modification. The samples were sprayed with aurum. The pressurized voltage was 15 kV and the resolution was 1.4 nm. TGA (Setaram Labsys Evo, France) was performed for thermal property characterization, where the samples were heated from 0

to 800°C under an N$_2$ flow at a scanning rate of 10°C min$^{-1}$ XRD (BRUKER AXSLTD, Germany) was used to detect the crystal structure. The fibre sample was fully ground into a powder. The determination conditions were as follows: Cu target, working voltage of 50 kV, current of 100 mA, scanning angle 2$\theta$ (Bragg angle) that varied from 5° to 60°, and scanning speed of 0.06° s$^{-1}$.

## 2.4. Antibacterial activity assay

*Escherichia coli* and *S. aureus* were used as the representative microorganisms in this study. The media used were 10 g l$^{-1}$ peptone, 5 g l$^{-1}$ beef extract, and 5 g l$^{-1}$ sodium chloride, at a pH of 7.2. After the bacteria were activated, 1 ml bacterial culture was centrifuged at 12 000 r.p.m. for 2 min, and the thallus was cleaned with sterile saline, added to 20 ml of sterile saline and dispersed uniformly. Subsequently, 0.2 g hemp fibre before and after modification was immersed into 20 ml bacterial suspension. The mixed solution was incubated at 37°C at 180 r.p.m. in a shaking incubator. After 24 h, 1 ml bacterial suspension was taken to make different dilutions of bacterial suspensions (from 10$^1$ to 10$^7$). Finally, 0.1 ml bacterial suspensions of 10$^5$, 10$^6$ and 10$^7$ were inoculated on the sterile nutrient broth agar medium. The plates were placed in an incubator at a culture temperature of 37°C for 24 h. The initial bacterial suspension (without fibre) was used as a negative control group. The number of living colonies was counted, and repeated thrice for each sample. The antibacterial ratio was quantified as follows:

$$\text{antibacterial ratio}(\%) = \frac{(A - B)}{A} \times 100, \tag{2.3}$$

where $A$ and $B$ are the number of colonies detected from the negative control group (without fibre) and the test group that contained fibre in contact with bacteria for 24 h, respectively.

# 3. Results and discussion

## 3.1. Preparation of hemp fibre grafted with quaternary ammonium groups

### 3.1.1. Alkalization

The hydroxyl group (–OH) in cellulose, hemicellulose and lignin formed a large number of hydrogen bonds between the macromolecules in the plant fibre cell wall [24]. To increase the availability of hydroxyl groups, NaOH was used during the alkalization process, because alkalization could break the intra-molecular or inter-molecular hydrogen bonds of cellulose, and at the same time, the ester bond between hemicellulose and coexisting components (lignin, pectin and lipids among others) will be saponified, and the ether bond will be broken under the action of OH$^-$, and part of the coexisting components will be dissolved in alkaline solution. Subsequently, the porosity and surface area of cellulose fibre will increase, whereas the crystallinity will decrease, which will improve the accessibility and activity of cellulose fibre [25,26]. The optimal NaOH concentration and alkalization time were determined according to the content of the aldehyde group formed by the hydroxyl group after the oxidation reaction. At different NaOH concentrations, 0.5 g hemp fibre was immersed in NaOH solution for 60 min at room temperature (20–30°C). The results (figure. 2$a$) show that the aldehyde group content increased with alkali concentration, and its maximum value (approx. 540.5 µmol g$^{-1}$) is observed at 15–18% NaOH solution. This was associated with the fact that the higher the alkali concentration (less than 15%), the more hydrogen bonds were eliminated. When the concentration was further increased (greater than 18%), the aldehyde group content significantly decreased, which was ascribed to the fact that the high concentration of alkali increased the metal ions in the solution to a certain extent, whereas the radius of hydrated ions decreased owing to the high concentration of ions, resulting in a decrease in swelling degree and the effect of alkalization [27]. For different reaction times, 0.5 g hemp fibre was immersed into 15% NaOH solution at room temperature (20–30°C). Figure 2$b$ shows that from the beginning to 60 min, the content of the aldehyde group increased, and the maximum value (557 ± 6.89 µmol g$^{-1}$) was obtained at 60 min for the 15% NaOH-treated fibre. It is worth mentioning that the raw hemp fibre was directly reacted with NaIO$_4$, and the content of aldehyde group generated was only 160 ± 5.64 µmol g$^{-1}$ because there were no more free hydroxyl groups. Therefore, it is critical to first treat hemp fibre with NaOH to increase the reactive hydroxyl groups. Figure 2 also shows the weight loss of the fibre during the alkalization reaction. The weight loss rate is affected by the alkali concentration and reaction time, but the

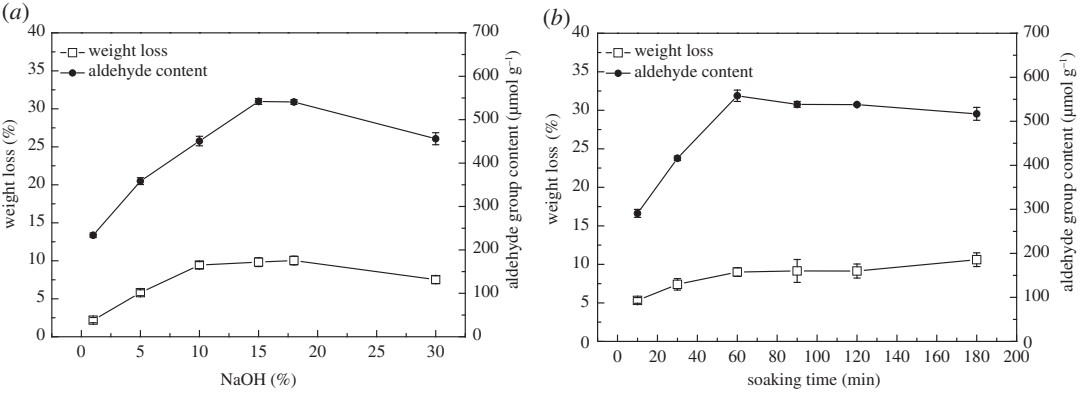

**Figure 2.** Effect of NaOH concentration (*a*) and soaking time (*b*) on the weight loss and aldehyde group content of alkalization reaction.

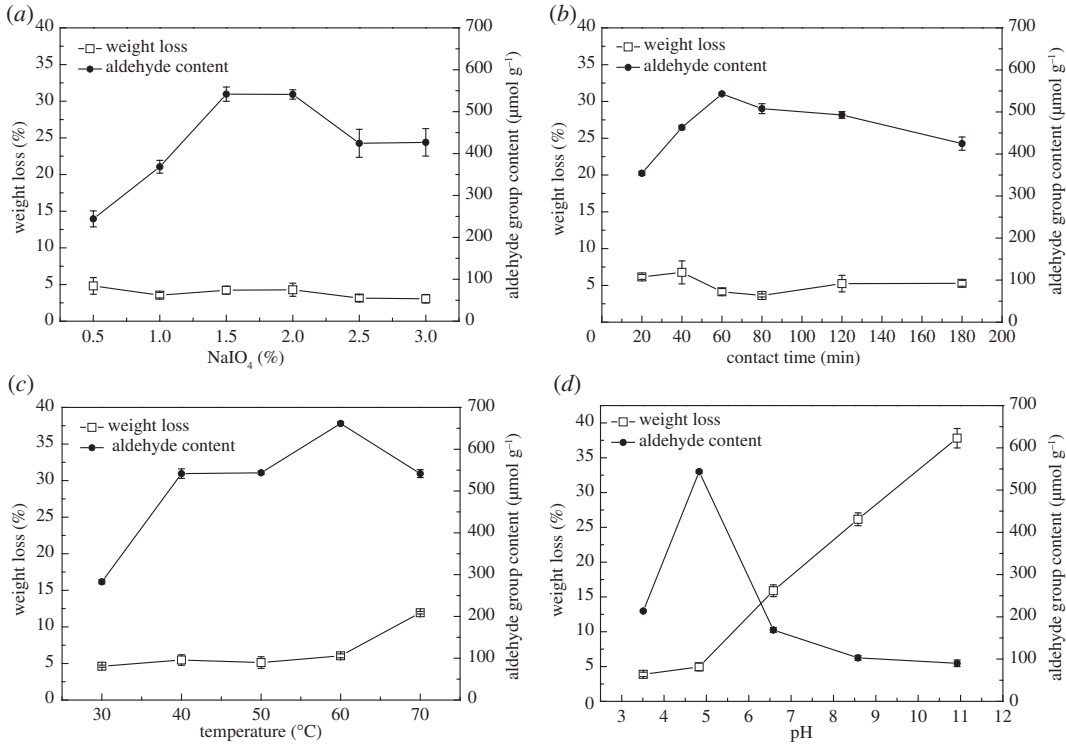

**Figure 3.** Effect of NaIO$_4$ concentration (*a*), contact time (*b*), temperature (*c*), and pH (*d*) on the weight loss and aldehyde group content of oxidation reaction.

influence is less (less than 10%), which is mainly caused by the dissolution of the coexisting components in the fibre and the shedding of shorter and thinner fibre filaments. The 15% NaOH treatment of hemp fibre for 60 min was the optimal condition, which was beneficial for producing more reactive functional groups for the next reaction, and the weight loss is acceptable.

### 3.1.2. Oxidation

The conversion of the hydroxyl group to the aldehyde group on the hemp fibre surface can provide chemical reaction sites for the Schiff base reaction. So far, the selective oxidation system for the secondary hydroxyl group is only periodate, which was discovered in 1937 [16]. Therefore, we choose NaIO$_4$ as the oxidant. Furthermore, the effects of NaIO$_4$ concentration, oxidation time, temperature, and pH on the aldehyde group content of HF–CHO were investigated. HF–OH (0.4 g) was reacted with NaIO$_4$ of different concentrations at 50°C in the dark at 180 r.p.m. for 60 min.

Figure 3a shows that the aldehyde group content increased with increasing NaIO$_4$ concentration. The maximum content was 541 ± 14 µmol g$^{-1}$ at 1.5% or 2% NaIO$_4$ with good fibrous morphology. When the concentration of NaIO$_4$ was over 2%, the aldehyde group content decreased, which may be owing to the fact that periodate gradually penetrates, diffuses and reacts from the amorphous region to the crystalline region of the hemp fibre during the reaction [28]. The macromolecules in the amorphous region are poorly arranged and the structure is loose, which is conducive to the penetration and reaction of the reagent. However, the macromolecules in the crystalline area are arranged neatly and densely, and the gap space in the fibre is small, which is not conducive to the penetration and diffusion reaction of the reagent, and the accessibility of the reagent to the fibre in the region is poor. On the contrary, in the presence of the high concentration of oxidant, the aldehyde group generated by oxidation will undergo an aldol condensation reaction with the unreacted hydroxyl group on the cellulose, resulting in a slight decrease in the aldehyde group content [29]. The concentration of NaIO$_4$ has no significant effect on the weight loss of hemp fibre within the range of the investigated conditions, but when the concentration of NaIO$_4$ exceeds 2%, the surface of the fibre will become stiff and lose the original fibrous morphology. Therefore, the appropriate concentration of NaIO$_4$ was determined as 1.5% to maintain a good fibre shape and low cost. For different reaction times, 0.4 g HF–OH was reacted with 1.5% NaIO$_4$ at 50°C in the dark at 180 r.p.m. Figure 3b shows that there was a sharp increase in the aldehyde group content from 20 min to 60 min, and the maximum content was 541 ± 3.06 µmol g$^{-1}$ at 60 min. Thereafter, the aldehyde group content slowly decreased until 120 min. With the prolongation of oxidation time, the degree of oxidation increased, and the content of the aldehyde group increased continuously. However, longer oxidation time (greater than 60 min) tends to a condensation reaction between the generated aldehyde group and the hydroxyl group of cellulose, resulting in the decrease of the aldehyde group content. The reaction time has no significant effect on the weight loss of fibre within the range of the investigated conditions, but the longer the reaction time, the stiffer the fibre. Therefore, the reaction time of 60 min was optimal. At different reaction temperatures, 0.4 g HF–OH was reacted with 1.5% NaIO$_4$ in the dark at 180 r.p.m. for 60 min. Figure 3c shows that a higher temperature enhanced the aldehyde group content of the final fibre. However, the reaction temperature of 70°C not only promoted the occurrence of the aldol condensation reaction, but also enhanced the oxidant penetration into the crystal area, resulting in a decrease in the aldehyde group content and the destruction of the crystal area of the fibre, thus increasing the weight loss of hemp fibre. Hemp fibre obtained the highest aldehyde group content at 60°C, but its morphology was not as good as that at low temperature because of the high oxidation degree. Therefore, a temperature of 40°C was considered to be the most favourable reaction temperature. At different pH values, 0.4 g HF–OH was reacted with 1.5% NaIO$_4$ in the dark at 180 r.p.m. for 60 min. Figure 3d shows that at a pH of 4.83, which was the pH of the original NaIO$_4$ solution, the maximum content of the aldehyde group (543 ± 3.78 µmol g$^{-1}$) was received; other pH values were not conducive to the progress of oxidation, especially in alkaline conditions. Under alkaline conditions, the hydrated sodium ions in the solution will enter the crystalline area of the fibre, increasing its swelling degree [6], so that periodate is more likely to enter the crystalline area and destroy it, and ultimately cause a serious loss of fibre strength. That is, the weight loss of the fibre in figure 3d increases significantly with pH above 5. Of course, the loss of fibre structure must also be accompanied by a significant decrease in the aldehyde group content. Therefore, a pH of 4.83 (i.e. NaIO$_4$ directly dissolved in water) was a suitable value during the synthesis of HF–CHO. In short, NaIO$_4$ showed strong oxidation, and its oxidation became stronger with increasing concentration, temperature, and time. However, owing to excessive oxidation, the aldehyde group formed by hemp fibre reacts with the hydroxyl group, namely the aldol condensation reaction, resulting in severe dissolution and stiffness of hemp fibre. Therefore, controlling the degree of oxidation reaction provided a good matrix material for the next step. So, the optimal oxidation conditions were pH 4.83, 1.5% NaIO$_4$, heated to 40°C for 60 min, which was consistent with the reaction mechanism.

### 3.1.3. Amination

Figure 4a shows the effect of different solvents on the amination process. During the investigation, 0.25 ml TETA was dissolved in 40 ml of glycerol, propylene glycol, ethylene glycol, ethanol, acetone, distilled water, or 40 ml pure TETA, to which 0.3 g of HF–CHO fibre was added. The whole system was shaken at 160 r.p.m. for 6 h at 50°C, and during the reaction, 500 µl of 50% glutaraldehyde was added dropwise within 1 h, which was used as a cross-linking agent. The results showed that the fibre only maintained a good shape in acetone, where the weight gain was 17.21 ± 0.66%. In other

R. Soc. Open Sci. **8**: 201904

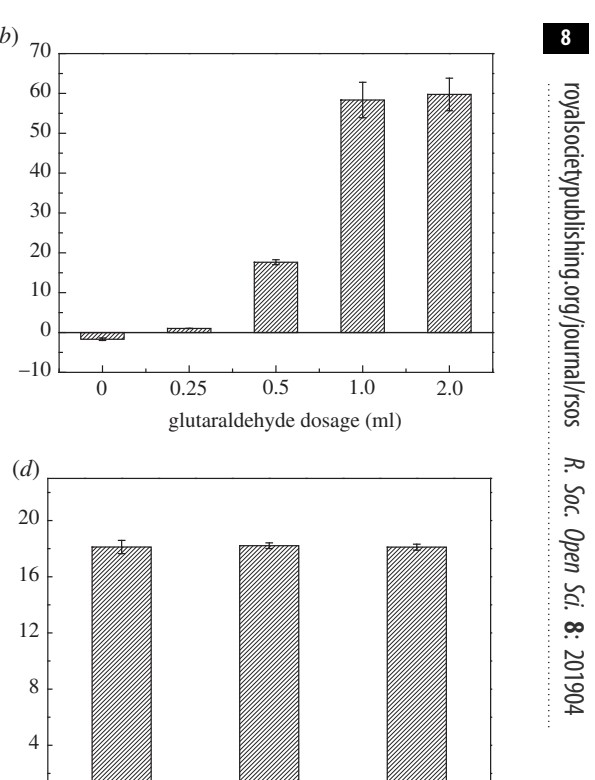

**Figure 4.** Effect of solvents (*a*), glutaraldehyde dosage (*b*), reaction time (*c*), and temperature (*d*) on the weight loss of amination reaction.

solvents, the weight gain was not significant or even negative, and the fibre was severely damaged (broke into fragments) after the reaction. Figure 4*b* demonstrates the effect of the glutaraldehyde dosage. During the investigation, 0.25 ml TETA was dissolved in 40 ml acetone, to which 0.3 g of HF–CHO fibre was added. The whole system was shaken at 160 r.p.m. for 6 h at 50°C, and during the reaction, different contents of 50% glutaraldehyde were added dropwise. From figure 4*b* it was clearly observed that glutaraldehyde had a positive effect on the amination. If glutaraldehyde was not added, it would have been difficult to obtain good fibre shape; a fibre with poor mechanical strength would have been obtained instead under normal conditions, and serious fibre loss would have occurred during washing. The higher the glutaraldehyde content in the reaction, the more significant the weight gain of the fibre. However, when the fibre weight increased by more than 50%, the fibre surface agglomerated seriously and clumps appeared, such that the fibrous morphology was destroyed. By contrast, 500 µl of 50% glutaraldehyde increased the fibre weight by 17.64 ± 0.61% and also maintained the original fibrous morphology. Figure 4*c* exhibits the effect of different contact times. During the investigation, 0.25 ml TETA was dissolved in 40 ml acetone, to which 0.3 g of HF–CHO fibre was added. The whole system was shaken at 160 r.p.m. for some time at 50°C, and during the reaction, 500 µl of 50% glutaraldehyde was added dropwise within 1 h. According to figure 4*c*, we could see that the fibre weight gain increased sharply at first and then decreased slowly, as the amination reaction time increased from 1 h to 12 h. If the reaction time was significantly long, the fibre would have been destroyed into fragments because of overreaction. Therefore, 6 h (corresponding to a weight gain of 18.43 ± 0.33%) was the most favourable reaction time. Figure 4*d* gives the effect of different reaction temperatures. During the investigation, 0.25 ml TETA was dissolved in 40 ml acetone, to which 0.3 g of HF–CHO fibre was added. The whole system was shaken at 160 r.p.m. for 6 h, and during the reaction, 500 µl of 50% glutaraldehyde was added dropwise within 1 h. Because the boiling point of acetone is 56.53°C, the temperature of the amination reaction could not be higher than it. The results indicated that when the temperature was 30, 40 or 50°C, there was no significant increase or decrease in the weight gain, proving that the temperature had no significant effect on the amination reaction. Therefore, the optimal condition for amination was that 0.3 g HF–CHO oxidized fibre was reacted with 0.25 ml TETA in acetone at 30–50°C for 6 h, and 500 µl of 50% glutaraldehyde was added dropwise within 1 h.

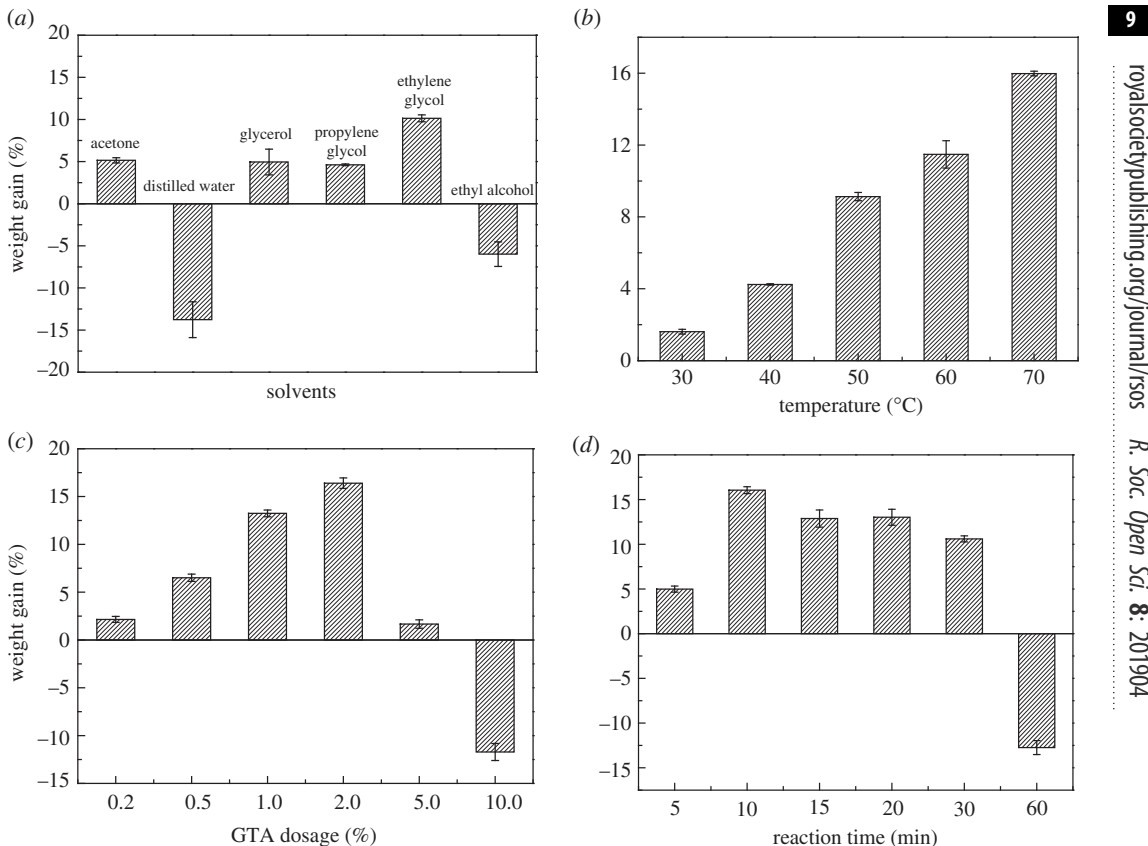

**Figure 5.** Effect of solvents (*a*), temperature (*b*), GTA dosage (*c*) and reaction time (*d*) on the weight loss of quaternization reaction.

### 3.1.4. Quaternization

During the quaternization reaction, quaternary ammonium groups were grafted onto the hemp fibre structure by N-alkylation between the amine and epoxy groups. In order to study the effect of different solvents on the quaternization reaction, GTA (2%) was reacted with 0.3 g of HF–NH$_2$ in different solvents at 50°C for 10 min, and the results are displayed in figure 5*a*. It showed that when water was used as the solvent, the fibre degraded significantly and the fibre morphology was seriously damaged and even turned into powder. When ethanol was used as the solvent, the fibre morphology was also damaged, resulting in fibre with a stiff surface. Fortunately, the quaternary ammonium groups could be grafted onto the fibre structure and resulted in a good morphology in acetone, propylene glycol, propylene glycol and ethylene glycol. Ethylene glycol produced the most significant weight gain, i.e. 10.14 ± 0.41%. Therefore, ethylene glycol was chosen as the optimal solvent for the quaternization reaction. For investigating the effect of reaction temperature on quaternization, GTA (2%) was reacted with 0.3 g of HF–NH$_2$ in 30 ml ethylene glycol for 10 min, and the results are shown in figure 5*b*. From it we show that when the temperature was higher than 70°C, the reaction rate of the fibre was very fast, which led to an uneven fibre surface and destroyed the morphology, which was difficult to control. Therefore, the highest temperature was set at 70°C. In addition, the weight gain increased significantly with the temperature and the maximum weight gain (15.98 ± 0.13%) was obtained at 70°C. Therefore, the optimal temperature of the quaternization reaction was 70°C. The influence of GTA dosage on quaternization was investigated, as shown in figure 5*c*. During the investigation, different contents of GTA were reacted with 0.3 g of HF–NH$_2$ in 30 ml ethylene glycol for 10 min at 70°C. The results showed that with the increase of GTA concentration, the fibre weight gain first increased and then decreased. When the concentration of GTA was 10%, the side reaction resulted in the degradation of the fibre into powder in large quantities. The optimal weight gain reached 16.4 ± 0.55% when the concentration of GTA was 2%. Finally, the effect of reaction time on quaternization was investigated, and the results are shown in figure 5*d*. At different reaction times, 2% of GTA was reacted with 0.3 g of HF–NH$_2$ in 30 ml ethylene glycol at 70°C. The results indicated that with the increase of time, the fibre weight gain first increased and then decreased. The longer the reaction time was, the more unfavourable the

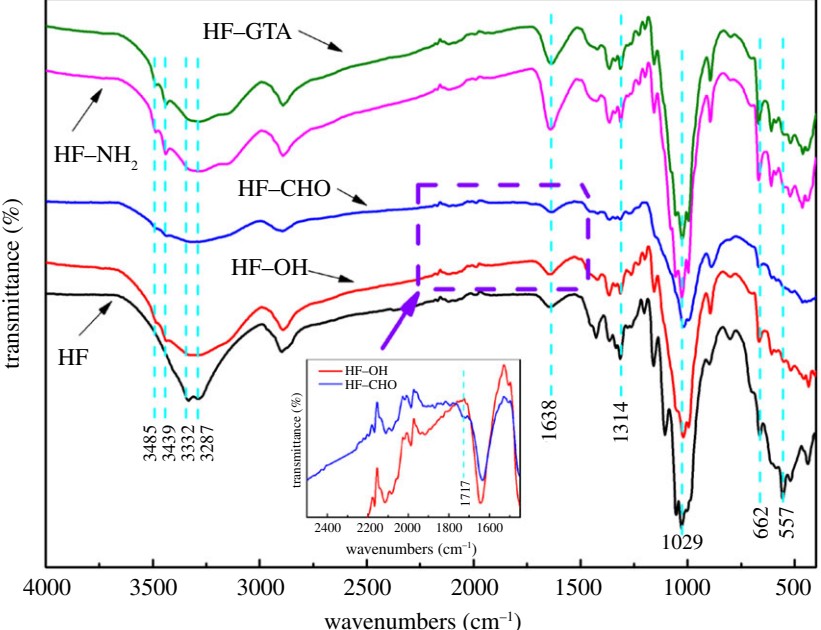

**Figure 6.** FTIR spectra of HF, HF–OH, HF–CHO, HF–NH$_2$ and HF–GTA.

quaternization reaction. The fibre weight gain reached the maximum at only 10 min, which was 16.05 ± 0.38%. Concurrently, the quaternary ammonium groups could be steadily grafted onto the fibre structure. Therefore, the optimal condition for quaternization was the immersion of 0.3 g of HF–NH$_2$ in 30 ml of 2% GTA ethylene glycol solution and reacted for 10 min at 70°C.

## 3.2. Characterization

The FTIR spectra of HF, HF–OH, HF–CHO, HF–NH$_2$ and HF–GTA are shown in figure 6. The broad band between 3000 cm$^{-1}$ and 3600 cm$^{-1}$ was ascribed to the adsorption by hydroxyl groups [30]. The peaks at 3287 cm$^{-1}$ and 3332 cm$^{-1}$ are ascribed to the stretching vibration of hydroxyl groups with intra-molecular hydrogen bonds and inter-molecular hydrogen bonds, respectively [31,32]. After alkalization, the peaks at these positions become wider, and two small new peaks appear at 3439 cm$^{-1}$ and 3485 cm$^{-1}$, which are ascribed to the destruction of hydrogen bonds in hemp fibre and the removal of non-cellulosic components which was earlier demonstrated in the weight loss after the alkalization [24,30], as well as the introduction of bound water into the fibre during the reaction process. Furthermore, after the sodium periodate treatment, the intensity of this band decreased owing to the conversion of a hydroxyl group to an aldehyde group [33]. The peak at 1638 cm$^{-1}$ is owing to the stretching vibration of C=C, N–H or C=N. After amination, it is significantly enhanced, which is owing to the superposition of C=N stretching vibration, N–H deformation vibration, and C=C stretching vibration. The peak was slightly weakened after quaternization, which also proved that the quaternization reaction occurred on N–H. HF–CHO has a weak peak at 1717 cm$^{-1}$, which is caused by the stretching vibration of the aldehyde group formed by oxidation [34]. The asymmetric and symmetric stretching vibrations of the C–O bonds (C–O–C and C–OH) and the substituted phenyl C–H can be observed in the region of 1185–837 cm$^{-1}$ (1160, 1105, 1053, 1029, 1001, 985 and 897 cm$^{-1}$) [32]. The infrared spectra of the five fibres proved that the chemical grafting reaction was successfully completed, and the reaction mainly occurred on the hydroxyl groups of hemp fibre.

The microscopic surface morphology of hemp fibre is shown in figure 7. The surface morphology of parent hemp fibre was relatively smooth. After alkalization and oxidation, the gullies on the surface of fibre deepened significantly. The surface of the fibre changed significantly and several protrusions appeared, which were primarily caused by the grafting reaction of the amine group in the amination reaction. In quaternization, the protrusions became bigger, indicating that modification caused the surface of the fibre to become rough, which could have reduced the mechanical property of the fibre. Therefore, suitable reaction conditions should be considered for production of better fibre with a high density of target functional groups and satisfactory fibrous configuration, which could also help to

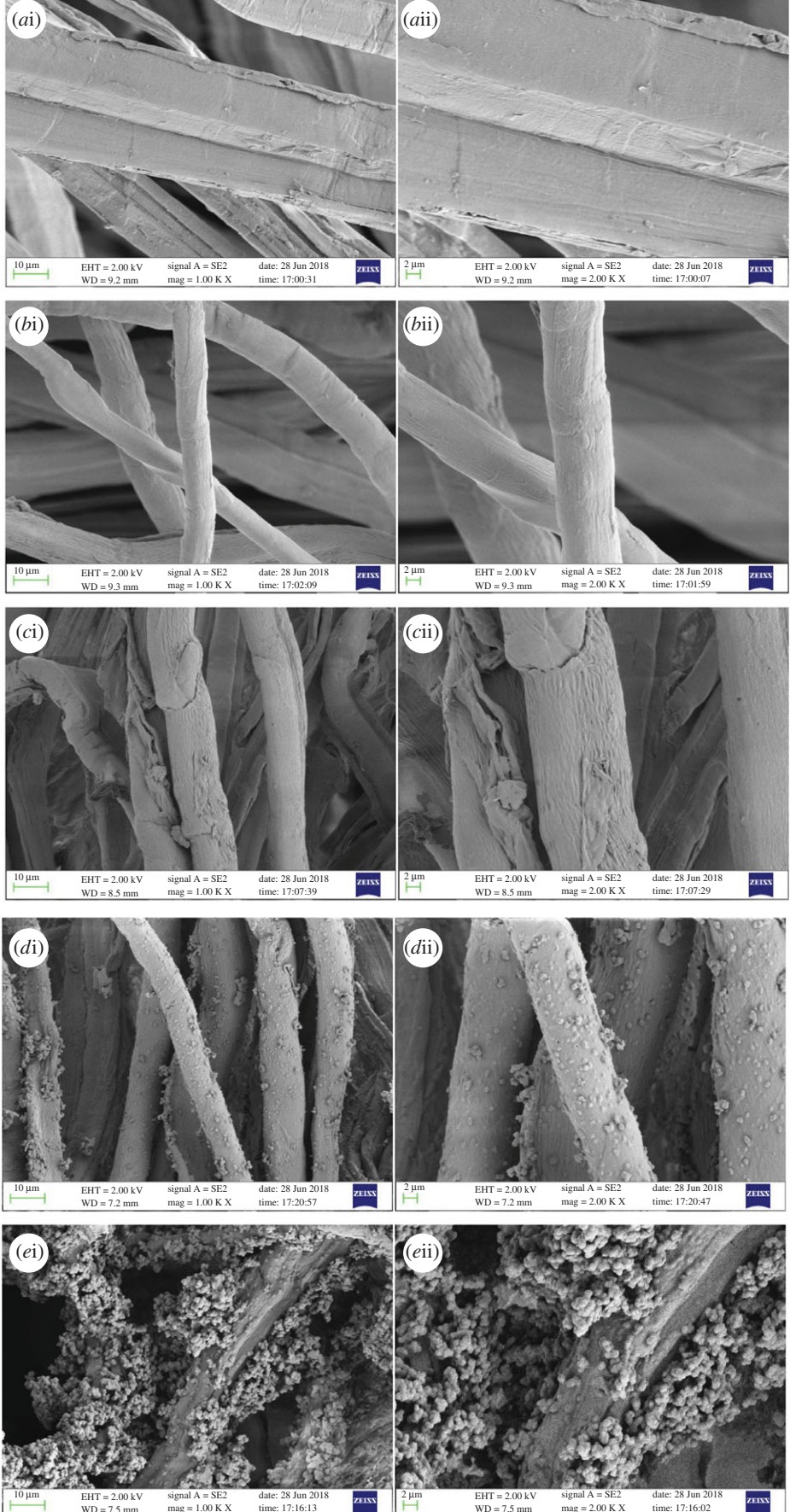

**Figure 7.** SEM images of HF (*a*), HF−OH (*b*), HF−CHO (*c*), HF−NH$_2$ (*d*), and HF−GTA (*e*).

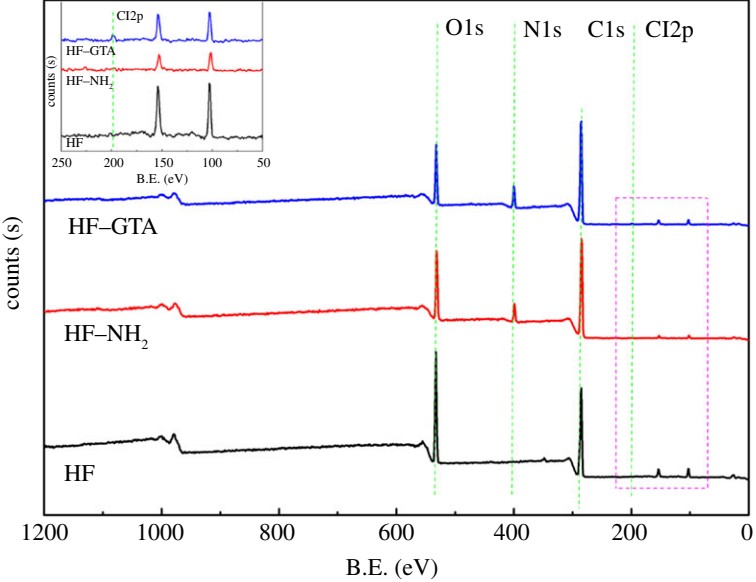

**Figure 8.** XPS survey scans of HF, HF–NH$_2$ and HF–GTA.

**Table 1.** The XPS analysis of HF, HF–NH$_2$ and HF–GTA.

| name | HF peak B.E. (eV) | atomic % | HF–NH$_2$ peak B.E. (eV) | atomic % | HF–GTA peak B.E. (eV) | atomic % |
|---|---|---|---|---|---|---|
| C1s | 284.82 | 70.38 | 284.78 | 72.41 | 284.83 | 74.27 |
| N1s | 400.03 | 0.79 | 399.35 | 8.23 | 399.25 | 9.54 |
| O1s | 532.59 | 28.83 | 531.99 | 19.35 | 531.96 | 15.93 |
| Cl2p | | | | | 197.01 | 0.25 |

increase the surface area of the fibre and promote the modification process. Although it was clear that GTA was grafted onto hemp fibre throughout the four steps, from the surface morphologies of fibres at each step, we found that the third and fourth steps were very important. Improper reaction conditions led to uneven grafting and overreaction on the fibre surface, which significantly affected the mechanical property of hemp fibre, which was consistent with the actual preparation procedure.

Furthermore, combined with the survey scan (figure 8) and peak table from XPS (table 1), it can be observed that the nitrogen content (at%) increased sharply from HF to HF–HN$_2$ and gently from HF–HN$_2$ to HF–GTA, which was consistent with the grafting processes; and the peak of chloride (0.25 at %) was also detected by XPS for HF–GTA. These results further confirmed the successful grafting of quaternary ammonium functional groups onto the structure of hemp fibre.

XRD was employed to study the changes in the crystal structure of hemp fibre before and after modification, and the results are shown in figure 9. For hemp fibre, there were three diffraction peaks at approximately 14.9° (1–10), 16.3° (110), and 22.6° (200), which were ascribed to the cellulose I$_\beta$ crystalline zone [35,36]. After alkalization the diffraction peaks at 16.3° became weaker, simultaneously, there grew a weak shoulder peak on the diffraction peak of 22.6°, which was caused by the transformation of cellulose I$_\beta$ to cellulose II during the alkalization. However, the conversion was not extensive. This phenomenon was also reported by Klemm $et\ al.$ [37], who observed that phase transition occurred within the regions of the crystalline order when lye concentrations exceeded 12–15%. This transformation is attributed to the intracrystalline swelling caused by the inclusion of NaOH and H$_2$O into the crystallites. This was also confirmed by the differential scanning calorimetry (DSC) analysis of HF and HF–OH (figure 10$b$), in which the peak temperature of the melting endothermic peak was nearly the same. In addition, the crystallinity of HF and HF–OH was calculated using equation (3.1) [38] to be 75.63% and 73.97%, respectively, implying that the crystallinity of HF–OH decreased. However, the crystal structure of the fibre was completely altered by the oxidation reaction, and the crystal structure

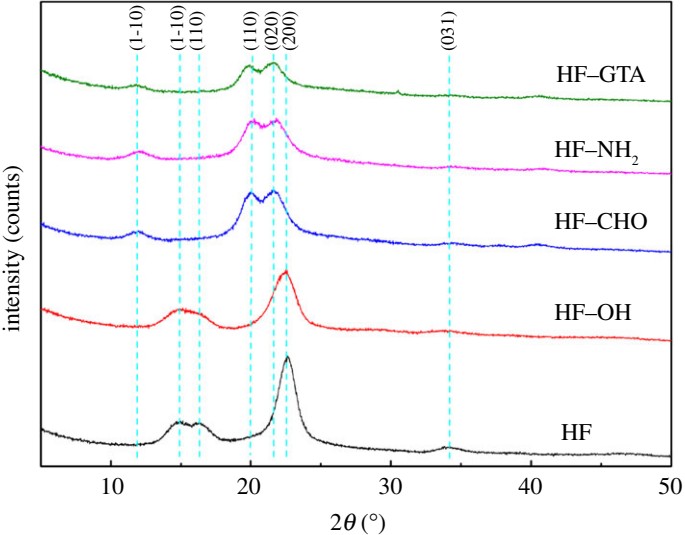

**Figure 9.** XRD curves of HF, HF–OH, HF–CHO, HF–NH$_2$ and HF–GTA.

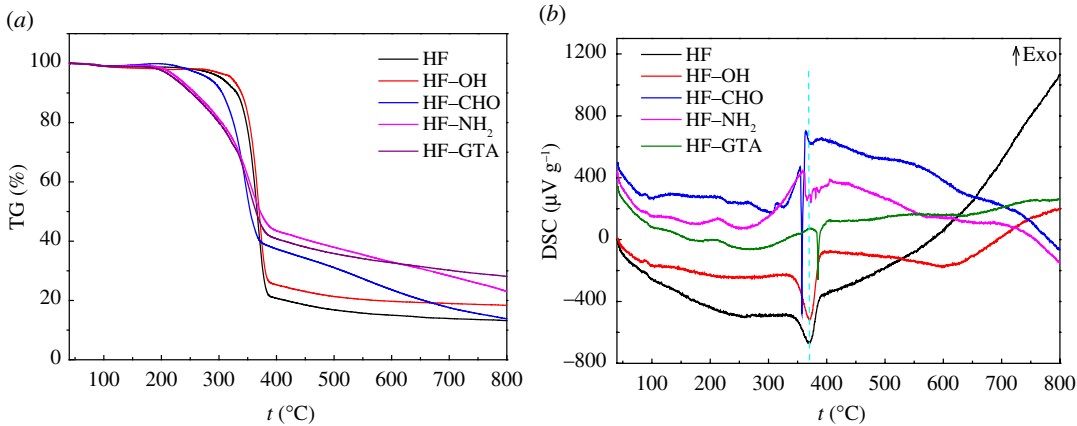

**Figure 10.** TGA (*a*) and DSC (*b*) curves of HF, HF–OH, HF–CHO, HF–NH$_2$ and HF–GTA.

conformed to the cellulose II crystalline zone. Moreover, the three main peaks at 11.82°, 19.92° and 21.61° corresponded to the Miller indices of (1–10), (110) and (020). The crystallinity of HF–CHO decreased to 67.17%. The crystal structure of HF–NH$_2$ and HF–GTA was similar to that of HF–CHO, but the crystallinity decreased to 60.48% and 60.35% after amination and quaternization.

$$I_{Cr} = \frac{I_{200} - I_{am}}{I_{200}} \times 100\%, \tag{3.1}$$

where $I_{Cr}$ is the crystallinity, $I_{200}$ is the diffraction peak intensity of the crystal plane (200), and $I_{am}$ is the intensity of the diffraction peak in the amorphous region (2$\theta$ = 19.0° for cellulose I).

From the TGA curves of HF and HF–OH in figure 10*a*, it can be observed that these fibres retained similar thermal properties, with only slight changes observed. The degradation temperature was approximately 260–310°C, which was attributed to the thermal decomposition of hemicellulose and cellulose. This was ascribed to the equivalent crystal form of the two fibres. For HF–CHO, HF–NH$_2$, and HF–GTA, the diffraction peaks were approximately equal; however, compared with those of HF and HF–OH, the diffraction peaks changed significantly. In other words, the original crystal form of HF was completely changed, that is, the original structure of the fibre was completely changed. Although the latter three fibres exhibited the same crystal face from the XRD data, it can be observed from their DSC diagrams that their melting temperature significantly varied. This implied that their structural differences were exceedingly large, which was consistent with the experimental design. As can be observed from figure 10*a*, the thermal stability of the fibres significantly reduced owing to the structural changes caused by oxidation, amination, and quaternization, where the initial degradation

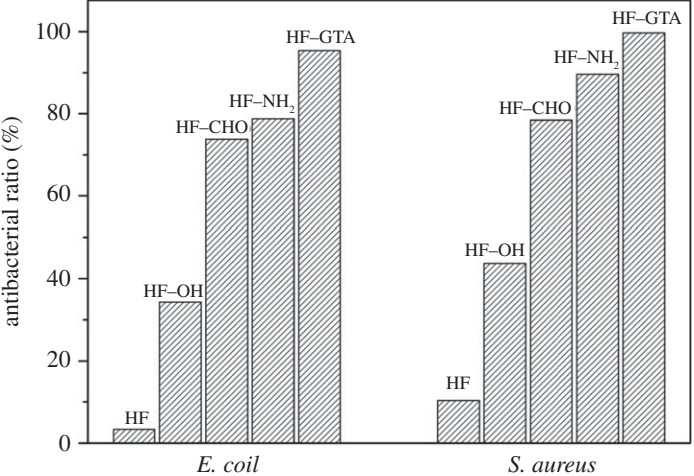

**Figure 11.** Antibacterial activities of HF, HF–OH, HF–CHO, HF–NH₂ and HF–GTA against pathogens after disinfecting 24 h.

temperatures were approximately 210°C, 185°C, and 185°C, respectively. In particular, for HF–NH$_2$ and HF–GTA, there were two degradation platforms at 185–330°C and 330–380°C, respectively. According to the literature [39], for hemp fibre, 260–330°C is the degradation platform of hemicellulose, and 330–380°C is the degradation platform of cellulose. Therefore, it can be concluded that amination and quaternization also occurred on the hemicellulose of hemp fibre. In addition, although the pyrolysis temperature of the modified hemp fibre decreased significantly, it maintained good thermal stability compared with its conventional-use environment. Furthermore, it can be observed from figure 10a that the five fibres demonstrated a very similar, small weight loss platform at approximately 130°C, which was caused by fibre absorbing moisture and small molecules from the air. It also showed that the modified hemp fibre retained the good hygroscopic property.

## 3.3. Antibacterial activity against pathogens

In this study, E. coli and S. aureus were used as the representative bacteria. It can be inferred from figure 11 that the fibres HF and HF–OH were not active against the pathogens, whereas the order of antibacterial activity of the other three fibres was as follows: HF–GTA > HF–NH$_2$ > HF–CHO. As expected, HF–GTA exhibited the highest antibacterial activity, and its antibacterial ratio reached more than 95%. The antibacterial activity of HF–CHO was in line with the results of previous studies, which proposed that the aldehyde group intrinsically endowed oxidized polysaccharides with broad-spectrum antibacterial activity [40,41]. For the antibacterial activity of HF–NH$_2$ and HF–GTA, the amine and quaternary ammonium groups had positive charges, which easily interacted with the negatively charged bacterial phospholipid outer membrane, resulting in changes in the permeability of the bacterial membrane, thereby causing cell membrane rupture, destruction, and dissolution [42–45]. Therefore, compared with the amine group, the quaternary ammonium group had a stronger positive charge, such that its antibacterial performance was stronger. Comparatively, HF–GTA exhibited lower antibacterial activity against E. coli (95.41%) than against S. aureus (99.64%) because of the differences in the cell membrane structure between Gram-negative bacteria and Gram-positive bacteria, which indicated that Gram-positive bacteria, such as S. aureus, carried more negative charges than Gram-negative bacteria, such as E. coli [45,46]. So, HF–GTA was less sensitive to E. coli. Considering our previous work, this novel antibacterial fibre HF–GTA showed better antibacterial activity than ethyltriphenylphosphonium bromide-polyacrylonitrile fibre [6]. We speculated that the hollow porous structure of hemp fibre increased the antibacterial effect when combined with quaternary ammonium groups. Coupled with the incomparable advantages of plant fibre, hemp fibre covalently grafted with quaternary ammonium groups are expected to become a more popular type of textile functional fibre.

## 3.4. Durability test of hemp fibre grafted with quaternary ammonium groups

The fibre HF–GTA was washed multiple times with 0.1% neutral detergent for 20 min, then rinsed with distilled water and dried overnight at 40°C. The washed fibre sample (0.2 g) was tested for antibacterial

**Table 2.** Washability of HF–GTA.

| number of washes | antibacterial ratio (%) | |
| --- | --- | --- |
| | *E. coli* | *S. aureus* |
| 0 | 95.41 | 99.64 |
| 10 | 93.31 | 96.67 |
| 30 | 89.78 | 91.12 |

activity according to §2.4. It is evident from the results shown in table 2 that the antibacterial ratios of HF–GTA exposed to *E. coli* and *S. aureus* for 24 h decreased with the increasing number of wash cycles. However, the antibacterial activity was retained even after washing 30 times. The antibacterial ratios of HF–GTA against *E. coli* and *S. aureus* remained 89.78% and 91.12%, respectively, indicating that HF–GTA was endowed with an appreciable washing resistance. Therefore, HF–GTA can be employed in the development of various antibacterial functional textiles and the sterilization of industrial and civil water.

# 4. Conclusion

Hemp fibre was grafted with quaternary ammonium groups (HF–GTA) for antibacterial application by alkalization, oxidation, amination, and quaternization multistage reactions, and the reaction conditions were optimized. The optimal process was as follows: firstly, hemp fibre was immersed in 15% NaOH solution for 60 min at room temperature (20–30°C); secondly, alkaline fibre was immersed in 1.5% $NaIO_4$ solution in the dark for 60 min at room temperature (20–30°C); thirdly, oxidized fibre was reacted with 0.25 ml TETA in acetone at 30–50°C for 6 h, and during the reaction, 500 µl of 50% glutaraldehyde was added dropwise within 1 h; finally, aminated fibre was reacted with GTA (2%) in propylene glycol at 70°C for 10 min. The chemical structure and micromorphology of hemp fibres were characterized by FTIR, XPS, SEM, TGA, and XRD. The successful grafting and reaction mechanism were proved, indicating that the grafting reaction primarily occurred on the hydroxyl groups of cellulose and hemicellulose in hemp fibre. The crystal form of the fibre changed from cellulose $I_\beta$ to cellulose II after grafting. Compared with the raw hemp fibre, the thermal stability of HF–GTA was greatly reduced, and the initial degradation temperature was reduced to 185°C. However, compared with the conventional application environment, it exhibited a good thermal stability and an excellent hygroscopicity. Furthermore, the antibacterial activities of hemp fibres before and after modification against *E. coli* and *S. aureus* were studied. The results indicated that HF–GTA exhibited the best antibacterial performance, where the antibacterial ratios against *E. coli* and *S. aureus* were 95.41% and 99.64%, respectively. Even after washing 30 times, the antibacterial activities were retained at 89.78% and 91.12%, respectively, indicating that HF–GTA was endowed with good washing resistance. The antibacterial activity of HF–GTA resulted from the positive charge carried by the quaternary ammonium group combined with the negative charge of the bacterial cell membrane, where the depletion of the electrochemical potential on the cell membrane led to the release of cytoplasmic substances and the dissolution of cells, thereby achieving antibacterial activity. Based on the excellent performance of HF–GTA, this work can provide support for the application of hemp fibre as a novel antibacterial material.

Data accessibility. Data available from the Dryad Digital Repository: https://doi.org/10.5061/dryad.z8w9ghx9m [47].

Authors' contributions. L.C. participated in all the experiments, did most of the data analysis and wrote the manuscript; W.D. analysed the data and revised the manuscript; A.C., J.L. and S.H. prepared the fibre and carried out the characterization; J.T., G.P. and Y.D. completed the antibacterial experiments; L.Z. and D.L. conceived the research and provided the necessary experimental equipment. All authors have given the final approval for publication.

Competing interests. We declare we have no competing interests.

Funding. This work was supported by the Natural Science Foundation of Hunan Province (grant no. 2018JJ3584) and China Agriculture Research System for Bast and Leaf Fibre Crops (grant no. CARS-16-E 02).

Acknowledgements. We are particularly grateful to the Key Laboratory of Microbial Engineering, Henan Academy of Sciences Institute of Biology, for supporting our antibacterial tests.

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
