## [Peer Review File · Royal Society Open Science]

Review History

RSOS-201904.R0 (Original submission)

Review form: Reviewer 1

Is the manuscript scientifically sound in its present form?

No

Are the interpretations and conclusions justified by the results?

Yes

Is the language acceptable?

Yes

Do you have any ethical concerns with this paper?

Yes

Have you any concerns about statistical analyses in this paper?

No

Recommendation?

Major revision is needed (please make suggestions in comments)

Comments to the Author(s)

The paper is well written and the presented results are interesting. However, there are many mistakes as well as too many unnecessary fibers characterizations such as TGA and DSC. Also, antimicrobial durability as well as fiber mechanical properties should be investigated. In my opinion, the paper is not acceptable for publication in the present form, major corrections should be made; some of them are listed below:

Abstract and Conclusion:

* "alkalinization", not "alkylation"

Introduction

* P1 L2: The word "corruption" is not appropriate in this context.

* P1 L19-20: Give some data (such as FAOSTAT) related to the sentence: "To date, China remains the chief producer of hemp fibers in the world"

Materials and Methods

*P3 L37: Did the authors determine the chemical composition of raw hemp fiber?

* P3 L39 and P4 L6: Please use one of these options: "2,3-Epoxypropyltrimethylammonium chloride" or "Glycidyl trimethyl ammonium chloride", not both.

* P3 L53: In section 3.2., there is no sufficient information regarding experimental conditions. It is better to give the alkalinization, oxidation, amination and quaternization conditions. For example, 1-30% NaOH, soaking time 5-180 min, not to say: "soaked in NaOH solution for some time". I know that these conditions are presented in figures 2-5, but in my opinion, it is better to give them also in 3.2. section. Moreover, the authors should write all the solvents used in amination and quaternization reactions, not to say "different solvents".

* P3 L59: Do the authors determine the weight loss (%) after each treatment (i.e. alkalinization, oxidation, amination and quaternization reaction)?

* P4 L: How do you determine the content of aldehyde groups? Describe the method in section 3.3.

Results and Discussion

* Through all manuscript and figures: "aldehyde group content", not "aldehyde content"

* P5 L14-L15: Can you describe the sentence: "To release more reactive hydroxyl groups, NaOH was used during alkanilization process". The hydroxyl groups were not released after the alkalinization, their availability was increased. Please establish the connection between the NaOH treatment and hemp non-cellulosic components. See the already cited reference 12 (Kostic et al. 2010) and the newest published literature from the same author such as:

Ivanovska, A., Asanovic, K., Jankoska, M. et al. Multifunctional jute fabrics obtained by different chemical modifications. *Cellulose* 27, 8485–8502 (2020). <https://doi.org/10.1007/s10570-020-03360-x>

* P5 L13-34: The discussion in this section is unclear. Try to better describe the results presented by using the following literature: D. Klemm, B. Philipp, T. Heinze, U. Heinze, W. Wagenknecht, "Comprehensive Cellulose Chemistry", Volume 1, Fundamentals and Analytical Methods, Wiley-VCH, Weinheim, 1998. (pp. 56 and 99).

* The content of aldehyde groups in raw hemp fibers should be given in Figs. 2 and 3.

*P5 L21: "up to 15%", not "below 15%"

* P5 L24-25: Support the statement "degradation of cellulose by the high-concentration alkali" by giving the results for cellulose degree of polymerization or some literature data.

- * P5 L31: During the treatment with NaOH the hydroxyl groups were not separated, their availability increased due to the removal of non-cellulosic components such as hemicelluloses, fats, waxes and pectin.
- * Description of Fig. 2b. Did the authors want to say: 20 ml 15% NaOH? or they further dilute 15% NaOH by adding 20 ml water? Make appropriate corrections in Fig. 3.
- * P5 L49-53: The authors should describe how the NaIO₄ concentration affected the aldehyde group content.
- * P7 L5: “providing that the temperature did not have significant effect...”, not “providing that contact time did not have significant effect...”
- * P8 L1-10: Omit the usage of the word “absorption” in the case of description of the FTIR transmittance spectra.
- * Please give the FTIR spectra of HF-OH and HF-CHO and describe the changes occurring due to the treatment with NaOH and NaIO₄.
- * P8 L32: “third and fourth steps”, not “thirdandfourthsteps”
- * From Fig. 8 it is evident that the highest content of nitrogen was detected in HF sample, while the lowest in HF-GTA sample. The presented results do not match with the discussion. Also, the chloride peak was not detected in the samples.
- * P8 L56- P9 34: Please use Cellulose 21:885-896 (2014) for the required nomenclature for X-ray diffraction peak labels and accordingly correct discussion in this section.
- * After the treatment with 15% NaOH, the structure of cellulose was changed, part of the cellulose I was converted to cellulose II. Please use Cellulose 21:885-896 (2014) for the required nomenclature for X-ray diffraction peak labels (for cellulose I and cellulose II) and accordingly correct discussion in this section.
- * The authors stated that after the alkali treatment, the “crystalline region became smaller”. How did the authors determine the fibers’ crystallinity?
- * Fig. 9. The changes in the XRD diffraction patterns were not clearly described.
- * Why the authors presented the TG and DSC analysis? In my opinion, these methods are not essential for characterizing the antimicrobial hemp fibers.
- * Having in mind four successive chemical reactions, the authors should determine the hemp fibers’ mechanical properties. If the treated hemp fibers have not good mechanical properties, what will be their end-use?
- * In many cases, washing durability is a key parameter for assessing the performance of antimicrobial fibers. How much of the antimicrobial effect survives after repeated washings is the major issue addressed by most researchers. The authors should test the antimicrobial durability of chemically treated hemp fibers?

Abstract and Conclusion:

Rewrite the abstract according to the given suggestions in the section “Results and discussion”.

Review form: Reviewer 2

Is the manuscript scientifically sound in its present form?

Yes

Are the interpretations and conclusions justified by the results?

Yes

Is the language acceptable?

Yes

Do you have any ethical concerns with this paper?

Yes

Have you any concerns about statistical analyses in this paper?

No

Recommendation?

Major revision is needed (please make suggestions in comments)

Comments to the Author(s)

The results are interesting since new antibacterial materials need to be investigated for the actual pandemic, but it needs a major revision. Check attached file (Appendix A)!

Decision letter (RSOS-201904.R0)

Dear Dr Chang:

Title: Improved antibacterial activity of hemp fibres by covalent grafting of quaternary ammonium groups

Manuscript ID: RSOS-201904

The editor assigned to your manuscript has now received comments from reviewers. We would like you to revise your paper in accordance with the referee and Subject Editor suggestions which can be found below (not including confidential reports to the Editor). Please note this decision does not guarantee eventual acceptance.

Please submit your revised paper before 25-Dec-2020. Please note that the revision deadline will expire at 00.00am on this date. If we do not hear from you within this time then it will be assumed that the paper has been withdrawn. In exceptional circumstances, extensions may be possible if agreed with the Editorial Office in advance. We do not allow multiple rounds of revision so we urge you to make every effort to fully address all of the comments at this stage. If deemed necessary by the Editors, your manuscript will be sent back to one or more of the original reviewers for assessment. If the original reviewers are not available we may invite new reviewers.

When submitting your revised manuscript, you must respond to the comments made by the referees and upload a file "Response to Referees" in "Section 6 - File Upload". Please use this to document how you have responded to the comments, and the adjustments you have made. In

order to expedite the processing of the revised manuscript, please be as specific as possible in your response.

On behalf of the Subject Editor Professor Anthony Stace and the Associate Editor Professor Chaohua Cui.

RSC Associate Editor:
Comments to the Author:
(There are no comments.)

RSC Subject Editor:
Comments to the Author:
(There are no comments.)

Reviewers' Comments to Author:
Reviewer: 1

Comments to the Author(s)

The paper is well written and the presented results are interesting. However, there are many mistakes as well as too many unnecessary fibers characterizations such as TGA and DSC. Also, antimicrobial durability as well as fiber mechanical properties should be investigated. In my opinion, the paper is not acceptable for publication in the present form, major corrections should be made; some of them are listed below:

Abstract and Conclusion:
* "alkalinization", not "alkylation"

Introduction

* P1 L2: The word "corruption" is not appropriate in this context.
* P1 L19-20: Give some data (such as FAOSTAT) related to the sentence: "To date, China remains the chief producer of hemp fibers in the world"

Materials and Methods

*P3 L37: Did the authors determine the chemical composition of raw hemp fiber?

- * P3 L39 and P4 L6: Please use one of these options: “2,3-Epoxypropyltrimethylammonium chloride” or “Glycidyl trimethyl ammonium chloride”, not both.
- * P3 L53: In section 3.2., there is no sufficient information regarding experimental conditions. It is better to give the alkalization, oxidation, amination and quaternization conditions. For example, 1-30% NaOH, soaking time 5-180 min, not to say: “soaked in NaOH solution for some time”. I know that these conditions are presented in figures 2-5, but in my opinion, it is better to give them also in 3.2. section. Moreover, the authors should write all the solvents used in amination and quaternization reactions, not to say “different solvents”.
- * P3 L59: Do the authors determine the weight loss (%) after each treatment (i.e. alkalization, oxidation, amination and quaternization reaction)?
- * P4 L: How do you determine the content of aldehyde groups? Describe the method in section 3.3.

Results and Discussion

- * Through all manuscript and figures: “aldehyde group content”, not “aldehyde content”
- * P5 L14-L15: Can you describe the sentence: “To release more reactive hydroxyl groups, NaOH was used during alkanilization process”. The hydroxyl groups were not released after the alkalization, their availability was increased. Please establish the connection between the NaOH treatment and hemp non-cellulosic components. See the already cited reference 12 (Kostic et al. 2010) and the newest published literature from the same author such as: Ivanovska, A., Asanovic, K., Jankoska, M. et al. Multifunctional jute fabrics obtained by different chemical modifications. *Cellulose* 27, 8485-8502 (2020). <https://doi.org/10.1007/s10570-020-03360-x>
- * P5 L13-34: The discussion in this section is unclear. Try to better describe the results presented by using the following literature: D. Klemm, B. Philipp, T. Heinze, U. Heinze, W. Wagenknecht, "Comprehensive Cellulose Chemistry", Volume 1, Fundamentals and Analytical Methods, Wiley-VCH, Weinheim, 1998. (pp. 56 and 99).
- * The content of aldehyde groups in raw hemp fibers should be given in Figs. 2 and 3.
- * P5 L21: “up to 15%”, not “below 15%”
- * P5 L24-25: Support the statement “degradation of cellulose by the high-concentration alkali” by giving the results for cellulose degree of polymerization or some literature data.
- * P5 L31: During the treatment with NaOH the hydroxyl groups were not separated, their availability increased due to the removal of non-cellulosic components such as hemicelluloses, fats, waxes and pectin.
- * Description of Fig. 2b. Did the authors want to say: 20 ml 15% NaOH? or they further dilute 15% NaOH by adding 20 ml water? Make appropriate corrections in Fig. 3.
- * P5 L49-53: The authors should describe how the NaIO₄ concentration affected the aldehyde group content.
- * P7 L5: “providing that the temperature did not have significant effect...”, not “providing that contact time did not have significant effect...”
- * P8 L1-10: Omit the usage of the word “absorption” in the case of description of the FTIR transmittance spectra.
- * Please give the FTIR spectra of HF-OH and HF-CHO and describe the changes occurring due to the treatment with NaOH and NaIO₄.
- * P8 L32: “third and fourth steps”, not “thirdandfourthsteps”
- * From Fig. 8 it is evident that the highest content of nitrogen was detected in HF sample, while the lowest in HF-GTA sample. The presented results do not match with the discussion. Also, the chloride peak was not detected in the samples.
- * P8 L56- P9 34: Please use *Cellulose* 21:885-896 (2014) for the required nomenclature for X-ray diffraction peak labels and accordingly correct discussion in this section.
- * After the treatment with 15% NaOH, the structure of cellulose was changed, part of the cellulose I was converted to cellulose II. Please use *Cellulose* 21:885-896 (2014) for the required

nomenclature for X-ray diffraction peak labels (for cellulose I and cellulose II) and accordingly correct discussion in this section.

* The authors stated that after the alkali treatment, the “crystalline region became smaller”. How did the authors determine the fibers’ crystallinity?

* Fig. 9. The changes in the XRD diffraction patterns were not clearly described.

* Why the authors presented the TG and DSC analysis? In my opinion, these methods are not essential for characterizing the antimicrobial hemp fibers.

* Having in mind four successive chemical reactions, the authors should determine the hemp fibers’ mechanical properties. If the treated hemp fibers have not good mechanical properties, what will be their end-use?

* In many cases, washing durability is a key parameter for assessing the performance of antimicrobial fibers. How much of the antimicrobial effect survives after repeated washings is the major issue addressed by most researchers. The authors should test the antimicrobial durability of chemically treated hemp fibers?

Abstract and Conclusion:

Rewrite the abstract according to the given suggestions in the section “Results and discussion”.

Reviewer: 2

Comments to the Author(s)

The results are interesting since new antibacterial materials need to be investigated for the actual pandemic, but it needs a major revision. Check attached file!

Author's Response to Decision Letter for (RSOS-201904.R0)

See Appendices B & C.

RSOS-201904.R1 (Revision)

Review form: Reviewer 1

Is the manuscript scientifically sound in its present form?

Yes

Are the interpretations and conclusions justified by the results?

Yes

Is the language acceptable?

Yes

Do you have any ethical concerns with this paper?

No

Have you any concerns about statistical analyses in this paper?

No

Recommendation?

Accept with minor revision (please list in comments)

Comments to the Author(s)

The quality of the manuscript is improved, however, some additional corrections should be made.

P3 L2: Please insert the chemical composition of hemp fibers obtained by the company.

P3 L17: As I can see in Fig. 3d, the pH range is from 3.5 to 11, not from 3 to 11.

P3: Rewrite the method for the determination of aldehyde group content. For example: "3-5 drops of thyme blue indicator were added to 50 mL of 20 g/L hydroxylamine hydrochloride methanol solution", not "take 50 mL of 20 g/L hydroxylamine hydrochloride methanol solution and add 3-5 drops of thyme blue as indicator." Moreover, check the indicator name, I am not sure that it is thyme blue. Give some reference where the mentioned procedure was used.

The captions of figures 2, 3, 4 and 5 are so long, rewrite them.

Fig. 2 and 3: "aldehyde group content", not "aldehyde content"

P4 L26, L54: The authors did not understand my comment. The hydroxyl groups were not released or produced, their availability increased after the treatment with sodium hydroxide. Namely, hydroxyl groups are already present in the fibers, and as a result of sodium hydroxide treatment (which selectively removed hemicelluloses), their availability increased.

P4 L32-38: The authors should give some reference/s.

P4 L38: This sentence is incorrect: "Notably, aldehyde groups are absent in raw hemp fibres." The aldehyde groups are present in the lignin. Since you have about 3-7% lignin, I am sure that the raw hemp fibers have a small amount of aldehyde groups. Please determine it and present the results in the figures. The small amount of aldehyde groups were also found in other bast fibers such as flax and jute. Check the following literature:

Milanovic et al. 2012 Influence of TEMPO-Mediated Oxidation on Properties of Hemp Fibers

Ivanovska et al. 2020 Waste Jute Fabric as a Biosorbent for Heavy Metal Ions from Aqueous Solution

Lazic et al. 2017 Influence of hemicelluloses and lignin content on structure and sorption properties of flax fibers (*Linum usitatissimum* L.)

P4 L58: The weight loss obtained after the sodium hydroxide treatment is very high (between 15 and 30%). Why? Which are the coexisting components in the hemp fibers that are removed during the alkalization?

P5 Rewrite section 3.1.2. The authors describe all that we can see in the figures. However, they do not conclude why such results are obtained. Correlate experimental conditions with weight loss and aldehyde group content.

Section 3.2. Try to better describe the FTIR spectra of the samples. I can not fully agree with the authors. Namely, after the treatment with NaOH, the band between 3600 and 3000 cm⁻¹ was broadened due to the increased availability of cellulose hydroxyl groups (caused by the removal of non-cellulosic components, which was earlier proved by the weight loss after the alkalization). Additionally, after the sodium periodate treatment, the intensity of this band decreased since the hydroxyl groups were converted to aldehyde groups... Also, the authors did not give any reference/s in this section.

The authors choose to use "alkalinisation" instead of "alkalinization" throughout all manuscript. According to that, correct the Fig. 2 caption.

Review form: Reviewer 2

Is the manuscript scientifically sound in its present form?

Yes

Are the interpretations and conclusions justified by the results?

Yes

Is the language acceptable?

Yes

Do you have any ethical concerns with this paper?

No

Have you any concerns about statistical analyses in this paper?

No

Recommendation?

Accept as is

Comments to the Author(s)

The article has been corrected and adapted to the journal, I think it is ready to be accepted

Decision letter (RSOS-201904.R1)

Dear Dr Chang:

Title: Improved antibacterial activity of hemp fibres by covalent grafting of quaternary ammonium groups

Manuscript ID: RSOS-201904.R1

Thank you for submitting the above manuscript to Royal Society Open Science. On behalf of the Editors and the Royal Society of Chemistry, I am pleased to inform you that your manuscript will be accepted for publication in Royal Society Open Science subject to minor revision in accordance with the referee suggestions. Please find the reviewers' comments at the end of this email.

The reviewers and handling editors have recommended publication, but also suggest some minor revisions to your manuscript. Therefore, I invite you to respond to the comments and revise your manuscript.

Because the schedule for publication is very tight, it is a condition of publication that you submit the revised version of your manuscript before 05-Feb-2021. Please note that the revision deadline will expire at 00.00am on this date. If you do not think you will be able to meet this date please let me know immediately.

Kind regards,
Dr Laura Smith
Publishing Editor, Journals

On behalf of the Subject Editor Professor Anthony Stace and the Associate Editor Professor Chaohua Cui.

RSC Associate Editor:
Comments to the Author:
(There are no comments.)

RSC Subject Editor:
 Comments to the Author:
 (There are no comments.)

Reviewer comments to Author:
 Reviewer: 2

Comments to the Author(s)
 The article has been corrected and adapted to the journal, I think it is ready to be accepted

Reviewer: 1

Comments to the Author(s)
 The quality of the manuscript is improved, however, some additional corrections should be made.

P3 L2: Please insert the chemical composition of hemp fibers obtained by the company.

P3 L17: As I can see in Fig. 3d, the pH range is from 3.5 to 11, not from 3 to 11.

P3: Rewrite the method for the determination of aldehyde group content. For example: "3-5 drops of thyme blue indicator were added to 50 mL of 20 g/L hydroxylamine hydrochloride methanol solution", not "take 50 mL of 20 g/L hydroxylamine hydrochloride methanol solution and add 3-5 drops of thyme blue as indicator." Moreover, check the indicator name, I am not sure that it is thyme blue. Give some reference where the mentioned procedure was used.

The captions of figures 2, 3, 4 and 5 are so long, rewrite them.

Fig. 2 and 3: "aldehyde group content", not "aldehyde content"

P4 L26, L54: The authors did not understand my comment. The hydroxyl groups were not released or produced, their availability increased after the treatment with sodium hydroxide. Namely, hydroxyl groups are already present in the fibers, and as a result of sodium hydroxide treatment (which selectively removed hemicelluloses), their availability increased.

P4 L32-38: The authors should give some reference/s.

P4 L38: This sentence is incorrect: "Notably, aldehyde groups are absent in raw hemp fibres." The aldehyde groups are present in the lignin. Since you have about 3-7% lignin, I am sure that the raw hemp fibers have a small amount of aldehyde groups. Please determine it and present the results in the figures. The small amount of aldehyde groups were also found in other bast fibers such as flax and jute. Check the following literature:

Milanovic et al. 2012 Influence of TEMPO-Mediated Oxidation on Properties of Hemp Fibers

Ivanovska et al. 2020 Waste Jute Fabric as a Biosorbent for Heavy Metal Ions from Aqueous Solution

Lazic et al. 2017 Influence of hemicelluloses and lignin content on structure and sorption properties of flax fibers (*Linum usitatissimum* L.)

P4 L58: The weight loss obtained after the sodium hydroxide treatment is very high (between 15 and 30%). Why? Which are the coexisting components in the hemp fibers that are removed during the alkalization?

P5 Rewrite section 3.1.2. The authors describe all that we can see in the figures. However, they do not conclude why such results are obtained. Correlate experimental conditions with weight loss and aldehyde group content.

Section 3.2. Try to better describe the FTIR spectra of the samples. I can not fully agree with the authors. Namely, after the treatment with NaOH, the band between 3600 and 3000 cm⁻¹ was broadened due to the increased availability of cellulose hydroxyl groups (caused by the removal of non-cellulosic components, which was earlier proved by the weight loss after the alkalization). Additionally, after the sodium periodate treatment, the intensity of this band decreased since the

hydroxyl groups were converted to aldehyde groups... Also, the authors did not give any reference/s in this section.

The authors choose to use "alkalinisation" instead of "alkalinization" throughout all manuscript. According to that, correct the Fig. 2 caption.

Author's Response to Decision Letter for (RSOS-201904.R1)

See Appendix D.

RSOS-201904.R2 (Revision)

Review form: Reviewer 1

Is the manuscript scientifically sound in its present form?

No

Are the interpretations and conclusions justified by the results?

Yes

Is the language acceptable?

Yes

Do you have any ethical concerns with this paper?

No

Have you any concerns about statistical analyses in this paper?

No

Recommendation?

Accept with minor revision (please list in comments)

Comments to the Author(s)

The captions of the Figures 2, 3, 4 and 5 are not clear, rewrite them. For example: Figure 2. Effect of NaOH concentration (a) and soaking time (b) on the weight loss and aldehyde group content. Fig. 2 and 3: "aldehyde group content", not "aldehyde content". The authors change only the ordinates' captions. Correct the explanation within Figures 2 and 3.

The authors reported the following chemical composition of hemp fibers: (cellulose > 90%, hemicellulose 2-3%, lignin < 1%, pectin < 1%, lipids < 1%). According to that, the total content of non-cellulosic components is < 6%. However, the weight loss obtained after the sodium hydroxide treatment is very high (between 15 and 30%). Why?

There is a significant lack of scientific knowledge related to the hemp fibers' chemical composition and the effect of different chemical treatments on hemp fibers. The authors stated: "For shorter and thinner fibre filaments, in addition to the dissolution of pectin and other substances in the fibres, it will also cause the decomposition of the fibre itself, and disperse into the lye. So the weight loss rate of the fibre we measured is higher." ... First of all, hemp fibers are staple fibers, not filament fibers... Second, in the first revision, the authors confirmed that the

decomposition of cellulose after the sodium treatment was not occurred. Now, they say that due to the fiber decomposition, the weight loss after the alkali treatment occurs...

There are still exist three different forms: alkalisation, alkanisation and alkalization. Choose one of them and make appropriate corrections in the manuscript.

P9, L7: Remove one "peaks"

P9, L12: hemp molecules?!

In the section characterization, the authors reported on the influence of lignin dissolution on the peak intensity. If you have <1% lignin, it can not affect the peak intensity.

The sentence "If the solution is red, 0.03 mol/L sodium hydroxide methanol solution was added dropwise..." construction is not correct. English should be improved in the whole manuscript.

Decision letter (RSOS-201904.R2)

Dear Dr Chang:

Title: Improved antibacterial activity of hemp fibres by covalent grafting of quaternary ammonium groups

Manuscript ID: RSOS-201904.R2

Thank you for submitting the above manuscript to Royal Society Open Science. On behalf of the Editors and the Royal Society of Chemistry, I am pleased to inform you that your manuscript will be accepted for publication in Royal Society Open Science subject to minor revision in accordance with the referee suggestions. Please find the reviewers' comments at the end of this email.

The reviewers and handling editors have recommended publication, but also suggest some minor revisions to your manuscript. Therefore, I invite you to respond to the comments and revise your manuscript.

Because the schedule for publication is very tight, it is a condition of publication that you submit the revised version of your manuscript before 19-Feb-2021. Please note that the revision deadline will expire at 00.00am on this date. If you do not think you will be able to meet this date please let me know immediately.

Kind regards,
Dr Laura Smith
Publishing Editor, Journals

On behalf of the Subject Editor Professor Anthony Stace and the Associate Editor Professor Chaohua Cui.

RSC Associate Editor:
Comments to the Author:
(There are no comments.)

RSC Subject Editor:
Comments to the Author:
(There are no comments.)

Reviewer comments to Author:

Reviewer: 1

Comments to the Author(s)

The captions of the Figures 2, 3, 4 and 5 are not clear, rewrite them. For example: Figure 2. Effect of NaOH concentration (a) and soaking time (b) on the weight loss and aldehyde group content. Fig. 2 and 3: "aldehyde group content", not "aldehyde content". The authors change only the ordinates' captions. Correct the explanation within Figures 2 and 3.

The authors reported the following chemical composition of hemp fibers: (cellulose > 90%, hemicellulose 2-3%, lignin < 1%, pectin < 1%, lipids < 1%). According to that, the total content of non-cellulosic components is < 6%. However, the weight loss obtained after the sodium hydroxide treatment is very high (between 15 and 30%). Why?

There is a significant lack of scientific knowledge related to the hemp fibers' chemical composition and the effect of different chemical treatments on hemp fibers. The authors stated: "For shorter and thinner fibre filaments, in addition to the dissolution of pectin and other substances in the fibres, it will also cause the decomposition of the fibre itself, and disperse into the lye. So the weight loss rate of the fibre we measured is higher." ... First of all, hemp fibers are staple fibers, not filament fibers... Second, in the first revision, the authors confirmed that the decomposition of cellulose after the sodium treatment was not occurred. Now, they say that due to the fiber decomposition, the weight loss after the alkali treatment occurs...

There are still exist three different forms: alkalisation, alkalisation and alkalization. Choose one of them and make appropriate corrections in the manuscript.

P9, L7: Remove one "peaks"

P9, L12: hemp molecules?!

In the section characterization, the authors reported on the influence of lignin dissolution on the peak intensity. If you have <1% lignin, it can not affect the peak intensity.

The sentence "If the solution is red, 0.03 mol/L sodium hydroxide methanol solution was added dropwise..." construction is not correct. English should be improved in the whole manuscript.

Author's Response to Decision Letter for (RSOS-201904.R2)

See Appendix E.

RSOS-201904.R3 (Revision)

Review form: Reviewer 1

Is the manuscript scientifically sound in its present form?

Yes

Are the interpretations and conclusions justified by the results?

Yes

Is the language acceptable?

Yes

Do you have any ethical concerns with this paper?

No

Have you any concerns about statistical analyses in this paper?

No

Recommendation?

Accept as is

Comments to the Author(s)

I am satisfied with the authors response to my comments and recommend to accept the manuscript for publication.

Decision letter (RSOS-201904.R3)

Dear Dr Chang:

Title: Improved antibacterial activity of hemp fibre by covalent grafting of quaternary ammonium groups

Manuscript ID: RSOS-201904.R3

It is a pleasure to accept your manuscript in its current form for publication in Royal Society Open Science. The chemistry content of Royal Society Open Science is published in collaboration with the Royal Society of Chemistry.

On behalf of the Subject Editor Professor Anthony Stace and the Associate Editor Professor Chaohua Cui.

RSC Associate Editor:
Comments to the Author:
(There are no comments.)

RSC Subject Editor:
Comments to the Author:
(There are no comments.)

Reviewer(s)' Comments to Author:
Reviewer: 1

Comments to the Author(s)
I am satisfied with the authors response to my comments and recommend to accept the manuscript for publication.

Appendix A**ROYAL SOCIETY
OPEN SCIENCE****Improved antibacterial activity of hemp fibres by covalent
grafting of quaternary ammonium groups**

Journal:	Royal Society Open Science
Manuscript ID	RSOS-201904
Article Type:	Research
Date Submitted by the Author:	28-Oct-2020
Complete List of Authors:	Chang, Li; Chinese Academy of Agricultural Sciences Institute of Bast Fiber Crops, Duan, Wenjie; Henan Academy of Sciences Institute of Chemistry ; Zhengzhou University, School of Materials science and engineering Huang, Siqi; Chinese Academy of Agricultural Sciences Institute of Bast Fiber Crops Chen, Anguo; Chinese Academy of Agricultural Sciences Institute of Bast Fiber Crops Li, Jianjun; Chinese Academy of Agricultural Sciences Institute of Bast Fiber Crops Tang, Huijuan; Chinese Academy of Agricultural Sciences Institute of Bast Fiber Crops Pan, Gen; Chinese Academy of Agricultural Sciences Institute of Bast Fiber Crops Deng, Yong; Chinese Academy of Agricultural Sciences Institute of Bast Fiber Crops Zhao, Lining; Chinese Academy of Agricultural Sciences Institute of Bast Fiber Crops Li, Defang; Chinese Academy of Agricultural Sciences Institute of Bast Fiber Crops
Subject:	Materials science < CHEMISTRY
Keywords:	hemp fibres, quaternary ammonium, covalent grafting, antibacterial activity
Subject Category:	Chemistry

**Author-supplied statements**

Relevant information will appear here if provided.

**Ethics**

*Does your article include research that required ethical approval or permits?:*

This article does not present research with ethical considerations

*Statement (if applicable):*

CUST_IF_YES_ETHICS :No data available.

**Data**

*It is a condition of publication that data, code and materials supporting your paper are made publicly available. Does your paper present new data?:*

Yes

*Statement (if applicable):*

Our data are deposited at Dryad:doi.org/10.5061/dryad.z8w9ghx9m

For private access during the review period, we share our unpublished dataset using this temporary
link:https://datadryad.org/stash/share/aVIVsydADtqc50K8HJCnadZHpCACxSmmmcDuYBCnvJA.

**Conflict of interest**

I/We declare we have no competing interests

*Statement (if applicable):*

CUST_STATE_CONFLICT :No data available.

**Authors' contributions**

This paper has multiple authors and our individual contributions were as below

*Statement (if applicable):*

42 L.C. participated in all the experiments, did most of the data analysis and wrote the manuscript;
43 W.D. analyzed the data and revised the manuscript A.C., J.L. and S.H. prepared the fibres and
44 completed the characterization of the samples; J.T., G.P. and Y.D. completed the antibacterial
experiments; L.Z. and D.L. conceived the research and
provided the necessary experimental equipment. All authors gave final approval for publication.

Dr. Li Chang
Institute of Bast Fiber Crops,
Chinese Academy of Agricultural Sciences
Changsha, 410205, China
Phone / Fax: 086-731-88998531 (O)
Email: changli519@163.com
October 23, 2020

Editor, *Royal Society Open Science*

Dear Editor,

I wish to submit an article for publication in *Royal Society Open Science*, titled “Improved antibacterial activity of hemp fibres by covalent grafting of quaternary ammonium groups”.

This study aimed to synthesize novel, functional hemp fibres, which could be armed with quaternary ammonium groups for improved antibacterial activity by alkylation, oxidation, amination, and quaternization multistage reactions. The properties of the as-prepared fibres were characterized by Fourier-transform infrared spectroscopy, thermogravimetric analysis, scanning electron microscopy, X-ray photoelectron spectroscopy and X-ray diffraction. Moreover, their antimicrobial activities and mechanisms were examined. We believe that our study makes a significant contribution to the literature because it proposes a new strategy for the surface modification of hemp fibres. The grafting reaction primarily occurred on the hydroxyl group of cellulose and hemicellulose in the hemp fibre, such that the modified hemp fibre retained a good thermal stability and hygroscopicity. Our experiments show the reliability and stability of our products. In fact, our findings are supported by strongly relevant and well-known sources, and the multiple tests we conducted cooperated in demonstrating the principles of the fibre functionality. Our material, grafted with quaternary ammonium groups, showed remarkable antibacterial activity against representative gram-positive and gram-negative bacteria. Its antibacterial ratios against *E. coli* and *S. aureus* were 95.41% and 99.64%, respectively. Therefore, these fibres will act as self-sterilizing materials and can be used in many applications such as textiles, health care products, and hygienic applications.

This manuscript has not been published or presented elsewhere in part or in entirety and is not under consideration by another journal. We have read and understood your journal’s policies, and we believe that neither the manuscript nor the study violates any of these. There are no conflicts of interest to declare.

Thank you for your consideration. I look forward to hearing from you.

Sincerely,
Dr. Li Chang

Improved antibacterial activity of hemp fibres by covalent grafting of quaternary ammonium groups

Li Chang^a, Wenjie Duan^{bc}, Siqi Huang^a, Anguo Chen^a, Jianjun Li^a, Huijuan Tang^a, Gen Pan^a, Yong Deng^a, Lining Zhao^{a*}, Defang Li^{a*}

a. Institute of Bast Fibre Crops, Chinese Academy of Agricultural Sciences, 410205, Changsha, Hunan, China.

b. Institute of Chemistry, Henan Academy of Sciences, 450003, Zhengzhou, Henan, China. c. School of Materials Science and Engineering, Zhengzhou University, 450000, Zhengzhou, Henan, China.

Keywords: hemp fibres; quaternary ammonium; covalent grafting; antibacterial activity

1. Summary

The improvement in living standards has led to an increased demand for antibacterial fabrics. To meet these demands, hemp fibres were armed with quaternary ammonium to improve their antibacterial activity by alkylation, oxidation, amination, and quaternization multistage reactions, resulting in a novel material with good antibacterial performance. The antibacterial mechanism occurred because the electrostatic reaction reduced the electrochemical potential on the cell membrane, leading to the release of cytoplasmic substances and the dissolution of cells, thereby producing antibacterial activity. The chemical structure and micromorphology of the fibres were characterized by Fourier-transform infrared spectroscopy, X-ray photoelectron spectroscopy, scanning electron microscopy, thermogravimetric analysis, and X-ray diffraction. The grafting and reaction mechanisms were proved to be successful, which indicated that the grafting reaction primarily occurred on the hydroxyl group of cellulose and hemicellulose in the hemp fibre, where the modified hemp fibre retained good fibrous morphology, thermal stability and hygroscopicity. The antibacterial activity of the fibres against *E. coli* and *S. aureus* demonstrated that the quaternary ammonium hemp fibre exhibited the best antibacterial performance, where its antibacterial ratios against *E. coli* and *S. aureus* were 95.41% and 99.64%, respectively. It would be significantly important to guarantee textile quality and prevent disease transmission.

2. Introduction

Authors* for correspondence (csbtzn@163.com (Zhao L); 13873129468@126.com (Li D)).

†Present address: Institute of Bast Fibre Crops, Chinese Academy of Agricultural Sciences, Salt Lake West Road No. 348, Changsha, Hunan, 410205, China.

<https://mc.manuscriptcentral.com/rsos>

Harmful organisms are prolific in daily life. These organisms can not only lead to human and animal
diseases and death but also cause the decomposition, deterioration, and corruption of various materials. Textiles
are excellent habitats for these microorganisms and are important sources for disease spreading. The production
of textiles with antimicrobial properties is increasingly common and has become an area of significant research
focus. Finishing is a traditional method; however, the resultant textiles have poor antimicrobial durability. With
the development of science and technology, researchers have begun to use modification methods to produce
fibres with long-lasting antimicrobial effects. In modification methods, antibacterial agents are introduced into
the surface and the inside of the fibres by physical or chemical methods [1–6]. Because of the strong binding
between the antibacterial agents and fibres, the resulting antibacterial fibres maintain long-lasting antibacterial
effects.

Hemp (*Cannabis sativa* L.) is an annual herb, which has been dispersed and cultivated by humans across the
world, from the tropics to alpine foothills. It is one of the oldest plant sources for seed oil, intoxicant resin,
medicine, and even textile fibres, trailing back at least six millennia [7]. For a few millennia, hemp has been a
respected crop in China, where it became an important fibre for clothing [8,9]. To date, China remains the chief
producer of hemp fibres in the world. Because hemp has a low requirement for fertilizers and pesticides during
planting, the harvested fibre can be used as an excellent environmental textile fibre with rich resources. More
importantly, compared with other fibres, hemp fibres have specific properties, such as aseptic properties, high
absorbency and hygroscopicity, good thermal and electrostatic properties, protection against UV radiation, lack
of any allergenic effect, and low cost [9–12]. Furthermore, hemp fibres are very soft and their ends are blunt,
which prevents sharp itching. Nowadays, clothing made of hemp fibres is gradually becoming favourable, and
their market demand is accelerating [9]. According to recent studies, hemp bast fibres demonstrated some
antibacterial properties because of their hollow porous morphology and antibacterial substances that can
destroy the living environment of anaerobic bacteria [13]. Nevertheless, hemp bast fibres contain a large amount
of gum. Therefore, it is necessary to degum hemp bast fibres for textiles [14]. This process causes the loss of most
of the antibacterial components, resulting in the loss of the antibacterial properties of the hemp fibres; therefore,
it is difficult to meet the human demand for antibacterial hemp fibres.

The structure of hemp fibres comprises three major constituents: cellulose, hemicellulose, and lignin.
Cellulose is considered to be the major framework component of the fibre structure. The chemical structure of
cellulose contains three hydroxyl groups, which form hydrogen bonds in the macromolecular cellulose network.
Two of these hydroxyl groups form inter-molecular bonds, while the third forms intra-molecular hydrogen
bonds [15]. These hydroxyl groups can be used as reactive groups for a series of chemical reactions, such as
oxidation, esterification, etherification, swelling, and grafting reactions [16,17]. Therefore, hemp fibre is an ideal
raw material as an easily renewable source, with the potential for modification to prepare functional
antibacterial fibres. It is important to endow hemp fibres with antibacterial properties to produce an important
functional material that can guarantee textile quality and prevent disease transmission.

Quaternary ammonium salt is widely used as an efficient disinfectant in medical and health care, food,
daily chemicals, and other fields, and is currently the most studied organic antibacterial agent. Quaternary
ammonium salt contains cations and the cellulose surface usually contains anions; hence, it is easy to adsorb on
the surface. However, the antibacterial agents of quaternary ammonium salts are only adsorbed on the surface
of cellulose; hence, they are easily lost during use, thereby causing environmental pollution [18]. Therefore,
quaternary ammonium functional groups have been covalently grafted onto the surface of cellulose, thereby
greatly improving the antibacterial properties and stability. For example, Liu et al. [19] grafted three kinds of
quaternary ammonium salts onto a chitosan nanocomposite membrane via etherification that displayed
excellent biocidal abilities against both *S. aureus* and *E. coli*. Oyervides-Muñoz et al. [20] synthesised three
different ammonium salts with a carboxylic acid end group through a quaternization reaction and chemically

grafted them along the chitosan backbone using 1-ethyl-3-(3-dimethylaminopropyl) carbodiimide (EDC) as a carboxyl-activating agent for coupling with the primary amine groups, which significantly improved the antibacterial activity against *P. aeruginosa*. Xu et al. [21] prepared a transparent cellulose membrane grafted onto the different ratios of sulfobetaine methacrylate and [2-(Acryloyloxy)ethyl]trimethylammonium chloride via surface-initiated atom-transfer radical polymerisation. Due to the presence of quaternary ammonium salt groups, the antibacterial properties of the cellulose membrane were improved, where the best antibacterial rates were 95.1% against *S. aureus* and 90.5% against *E. coli*.

To date, the studies regarding the modification of hemp fibres for antibacterial properties have been rare. In this work, we report the novel preparation of functional, antibacterial hemp fibres by the covalent grafting of quaternary ammonium groups. The as-prepared hemp fibre modified by quaternary ammonium groups (HF-GTA) presented favourable antibacterial properties against gram-positive and gram-negative bacteria. The performance before and after the modification of the fibres was characterized by Fourier-transform infrared (FTIR) spectroscopy, X-ray photoelectron spectroscopy (XPS), scanning electron microscopy (SEM), X-ray diffraction (XRD), and thermogravimetric analysis (TGA). The obtained HF-GTA remained advantageously fibrous, which results in more resourceful models in practice. For example, these fibres could be used in filaments or staples or woven in thread and cloth. Otherwise, the fibres exhibited a much wider specific surface area than that of resins and granular materials, which results in an improved antibacterial activity.

3. Materials and Methods

3.1. Materials

Hemp fibres with a length of approximately 5 cm and count of 1318 Nm were obtained from Hunan Huasheng Group Co., Ltd., China. Sodium periodate (NaIO_4), triethylenetetramine (TETA), 2,3-Epoxypropyltrimethylammonium chloride (GTA), 50% glutaraldehyde, and hydroxylamine hydrochloride were purchased from Shanghai Macklin Biochemical Technology Co., Ltd., China. Sodium chloride, sodium hydroxide, glycerol, methanol, ethanol, ethylene glycol, propylene glycol, and acetone were obtained from Hunan Huihong Reagent Co., Ltd., China. Peptone and beef extracts were purchased from Beijing Solarbio Science & Technology Co., Ltd., China. Agar was obtained from BioFROXX Biotechnology Co., Ltd., China. All chemicals and reagents used were of analytical grade and used as received. Pathogenic *E. coli* and *S. aureus* were supplied by the Institute of Biology, Henan Academy of Sciences, Limited Liability Company, Zhengzhou, China. Distilled water was used for the experiments.

3.2. Preparation of hemp fibres grafted with quaternary ammonium groups

Four steps were used to prepare the antibacterial hemp fibres (HF-GTA). The first step was an alkalization reaction. The hemp fibre, used as the parent fibre, was soaked in NaOH solution for some time at room temperature (30 ± 2 °C), and the fibre was washed with deionised water until the pH became neutral. The obtained alkaline fibre HF-OH was dried at 40 °C to obtain a constant weight. Second, an oxidation reaction was performed. The HF-OH was reacted with NaIO_4 solution in the dark for some time at a certain temperature, then the fibre was immersed in 1% glycerol solution for 30 min to remove the unreacted NaIO_4 . The obtained

oxidized fibre, HF-CHO, was fully washed with deionised water and dried at 40 °C to obtain a constant weight. Third, an amination reaction was performed. The HF-CHO was reacted with triethylenetetramine (TETA) in different solvents and different temperatures for some time, then the aminated fibre, HF-NH₂, was washed with deionised water and dried at 40 °C to obtain a constant weight. Finally, a quaternization reaction was performed. The HF-NH₂ was reacted with (Glycidyl trimethyl ammonium chloride) GTA in different solvents and at different temperatures for some time, then the final, quaternized fibre, HF-GTA, was washed with deionised water and dried at 40 °C to obtain a constant weight. The resulting fibre was a hemp fibre with antibacterial functions. The reaction mechanism is shown in Figure 1.

Figure 1.

3.3. Characterization

FTIR (Nicolet-460, USA) spectra, between 400 and 4000 cm⁻¹, were collected. Samples were prepared in a KBr pellet. XPS spectra for the fibres, before and after modification, were analysed by an Escalab 250XiXPS with a monochromatic Al-Kα X-ray source, and an XPS peak 4.1 software was used to fit the calibrated, high-resolution spectra. SEM (JSM-7500F) was used to observe the surface morphology of the fibres before and after the reaction. The samples were sprayed with an aurum. The pressurised voltage was 15 kV and the resolution was 1.4 nm. TGA (Setaram Labsys Evo, France) was performed simultaneously for thermal property characterizations, where the samples were heated from 0 °C to 800 °C under an N₂ flow at a scanning rate of 10 °C/min. XRD (BRUKER AXSLTD, Germany) was used to detect the crystal structure. The fibre sample was fully ground into a powder. The determination conditions were as follows: a Cu target, working voltage of 50 kV, current of 100 mA, scanning angle 2θ (Bragg angle) that varied from 5° to 60°, and scanning speed of 0.06 °/s.

3.4. Antibacterial activity assay

E. coli and *S. aureus* were used as the representative microorganisms in this study. The media used were 10 g/L peptone, 5 g/L beef extract, and 5 g/L sodium chloride, at a pH of 7.2. After the bacteria were activated, 1 mL bacterial culture was centrifuged at 12 000 rpm for 2 min, and the thallus was cleaned with sterile saline, added to 20 mL of sterile saline, and dispersed uniformly. Subsequently, 0.2 g of either raw hemp fibres or modified hemp fibres was immersed into a 20 mL bacterial suspension. The mixed solution was incubated at 37 °C at 180 rpm in a shaking incubator. After 24 h, a 1 mL bacterial suspension was removed to make different dilutions of bacterial suspensions (from 10¹ to 10⁷). Finally, 0.1 mL bacterial suspensions of 10⁵, 10⁶, and 10⁷ were seeded onto a sterile nutrient broth agar. The plates were placed in an incubator at a culture temperature of 37 °C for 24 h. The initial bacterial suspension (without fibres) was used as a negative control group. The number of living colonies was counted, and repeated thrice for each sample. The inhibition of cell growth was quantified as follows:

$$\text{Antibacterial ratio (\%)} = \frac{(A - B)}{A} \times 100, \quad (1)$$

where A and B are the number of colonies detected from the negative control group (without fibres) and the test group that contained fibres in contact with bacteria for 24 h, respectively.

4. Results and Discussion

4.1. Preparation of hemp fibres grafted with quaternary ammonium groups

4.1.1. Alkalinization

The hydroxyl groups (-OH) in cellulose, hemicellulose, and lignin formed a large number of hydrogen bonds between the macromolecules in the plant fibre cell wall [22]. To release more reactive hydroxyl groups, NaOH was used during the alkalinization process. The optimal NaOH concentration and alkalinization time were determined according to the content of aldehyde groups formed by hydroxyl groups after the oxidation reaction. The results (Fig. 2a) showed that the aldehyde group content increased with alkali concentration, and the maximum value of aldehyde group content (approximate 540.5 $\mu\text{mol/g}$) was observed at 15–18% NaOH solution. This was associated with the fact that a higher alkali concentration (below 15%) resulted in an increased elimination of hydrogen bonds. When the concentration was further increased, the aldehyde group content significantly decreased, which was ascribed to the shedding of hydroxyl groups with the degradation of cellulose by the high-concentration alkali. Figure 2b shows the effect of different soaking times on the aldehyde content of the fibre. From the beginning to 60 min, the content of aldehyde groups increased, and the maximum value ($557 \pm 6.89 \mu\text{mol/g}$) was generated at 60 min for the 15% NaOH-treated fibre. It is worth mentioning that the raw hemp fibre was directly reacted with NaIO_4 , and the content of aldehyde groups generated was only $260 \pm 5.64 \mu\text{mol/g}$ because there were no more free hydroxyl groups. Therefore, it is critical to first treat hemp fibres with NaOH to separate more active hydroxyl groups. The 15% NaOH treatment of the fibres for 60 min was the optimal condition, which was beneficial for producing more reactive functional groups for the next reaction.

Figure 2.

4.1.2. Oxidation

The effect of the concentration of NaIO_4 , oxidation time, temperature, and pH on the aldehyde content of HF-CHO was investigated. Figure 3a shows the effect of the concentration of NaIO_4 . The HF-OH was reacted with different concentrations of NaIO_4 at 50 °C in the dark at 180 rpm for 60 min. When the concentration of NaIO_4 was 1.5% or 2.0%, an ideal aldehyde content of greater than $541 \pm 14 \mu\text{mol/g}$ was reached, with a good fibre shape. The proper concentration of NaIO_4 was determined to be 1.5% to maintain a good fibre shape and low cost. Figure 3b shows the effect of different contact times. When the concentration of NaIO_4 was 1.5%, there was a sharp increase in the aldehyde content from 10 min to 60 min, and the maximum value of the aldehyde content ($541 \pm 3.06 \mu\text{mol/g}$) was observed at 60 min. Thereafter, the aldehyde group content slowly decreased until 120 min. Therefore, a reaction time of 60 min was optimal. Figure 3c shows the effect of different reaction temperatures. A higher temperature enhanced the aldehyde content of the final fibre. When the reaction temperature was 70 °C, the aldehyde content decreased, such that the temperature at 70 °C was not conducive to oxidation. Although the fibre demonstrated a maximum aldehyde group content at 60 °C, the fibre became stiff. Therefore, a temperature of 40 °C was concluded to be the most favourable reaction temperature. Figure 3d

shows the effect of pH. At a pH of 4.83, which was the pH of the original NaIO_4 solution, the maximum value of aldehyde content ($543 \pm 3.78 \mu\text{mol/g}$) was observed. Other pH values were not conducive to the progress of oxidation, especially in the alkaline environment, which may be because the fibre continued to alkalise and hindered the progress of the oxidation reaction. Therefore, a pH of 4.83 (i.e., NaIO_4 directly dissolved in water) was a suitable value during the synthesis of HF-CHO.

In conclusion, the optimal synthesis conditions were a pH of 4.83, where the concentration of NaIO_4 was 1.5% when heated to 40°C for 60 min. This coincided with the reaction mechanism. During this reaction, NaIO_4 had a strong oxidation effect, and its oxidation became stronger with increasing concentration, temperature, and time. NaIO_4 reacted with the hemp fibre, which was gradually infiltrated, diffusing from the amorphous region to the crystalline region of cellulose. When the degree of oxidation was weak, NaIO_4 acted only in the amorphous region, and the structure of the fibre was insignificantly destroyed. As the degree of oxidation increased, NaIO_4 permeated from the amorphous region to the crystalline region. When the oxidation continued, the aldol condensation reaction occurred because of the aldehyde group formed by the oxidation of the hemp fibre and the hydroxyl group of cellulose, such that the hemp fibre was severely dissolved and became stiff. Therefore, controlling the degree of oxidation reaction provided a good matrix material for the next step.

Figure 3.

4.1.3. Amination

Figure 4a shows the effect of different solvents on the amination process. TETA (0.25 mL) was dissolved in 40 mL of glycerol, propylene glycol, ethylene glycol, ethanol, acetone, deionised water, or 40 mL pure TETA, to which 0.3 g of HF-CHO fibre was added. The whole system was shaken at 160 rpm for 6 h at 50°C and, during the reaction, 500 μL of 50% glutaraldehyde was added dropwise within 2 h, which was used as a crosslinking agent. The results showed that the fibre only maintained a good shape in acetone, where the weight gain was $17.21 \pm 0.66\%$. In other solvents, the weight gain was not significant or even negative, and the fibre was severely damaged (broke into fragments) after the reaction. Figure 4b shows the effect of the glutaraldehyde dosage. TETA (0.25 mL) was dissolved in 40 mL acetone, to which 0.3 g of HF-CHO fibre was added. The whole system was shaken at 160 rpm for 6 h at 50°C , and during the reaction, different contents of 50% glutaraldehyde were added dropwise. It was clearly observed that glutaraldehyde had a positive effect on the amination. If glutaraldehyde was not added during the reaction, it would have been difficult to obtain a good fibre shape; a fibre with poor mechanical strength under the normal conditions would have been obtained instead, and serious fibre loss would have occurred during washing. The greater the glutaraldehyde content in the reaction, the more significant the weight gain of the fibre HF- NH_2 . However, when the fibre weight increased by more than 50%, the fibre surface agglomerated seriously and clumps appeared, such that the fibrous form was destroyed. In contrast, 500 μL of 50% glutaraldehyde increased the fibre weight by $17.64 \pm 0.61\%$ and also maintained the original shape of the fibre. Therefore, in the amination reaction, 500 μL of 50% glutaraldehyde contributed to the graft formation of the amine groups. Figure 4c shows the effect of different contact times. As the amination reaction time increased from 1 h to 12 h, the fibre weight gain first increased sharply and then decreased slowly. If the reaction time was too long, the fibre would have been destroyed into fragments because of overreaction. Therefore, 6 h (corresponding to a weight gain of $18.43 \pm 0.33\%$) was the most favourable reaction time. Figure 4d shows the effect of different reaction temperatures on the weight gain. The boiling point of acetone is 56.53°C .

°C; hence, the temperature of the amination reaction could not be higher than this temperature. When the temperature was 30, 40, or 50 °C, there was no significant increase or decrease in weight gain, proving that the contact time did not have a significant effect on the weight gain for the amination reaction. Subsequently, 0.5 g of HF-NH₂ prepared under the optimal conditions was reacted with 2.5 wt% of GTA in 40 mL propylene glycol at 70 °C for 10 min. During the reaction, quaternary ammonium groups were produced by the N-alkylation between the epoxy and amine groups, and the weight gain was 17.21 ± 0.66%.

Figure 4.

4.1.4. Quaternization

The quaternization reaction occurred when the GTA was grafted onto the fibre by an N-alkylation reaction between the amino and epoxy groups, and a quaternary ammonium group was introduced into the hemp fibre structure. Figure 5a shows the effect of different solvents on the reaction. When water was used as the solvent, the fibre degraded significantly, and the morphology of the fibre was seriously damaged and even turned into powder. For ethanol as solvent, the morphology of the fibre was also damaged, and the resulting fibre had a stiff surface. Fortunately, the quaternary ammonium groups could be grafted onto the fibre structure and resulted in a good morphology in acetone, propylene glycol, propylene glycol, and ethylene glycol. Ethylene glycol produced the most significant weight gain, which was 10.14 ± 0.41%. Therefore, ethylene glycol was chosen as the optimal solvent for the quaternization reaction. The effect of reaction temperature on quaternization was investigated, and the results are shown in Figure 5b. When the temperature was higher than 70 °C, the reaction rate of the fibre was very fast, which led to an uneven fibre surface and destroyed the morphology, which was difficult to control. Therefore, the highest temperature was set at 70 °C. The weight gain increased significantly with the increase in temperature, where the maximum weight gain was 15.98 ± 0.13% at 70 °C. Therefore, the optimal temperature for the quaternary ammonium reaction was 70 °C. The influence of GTA dosage on quaternization was investigated, as shown in Figure 5c. With the increase in GTA concentration, the weight gain of the fibre first increased and then decreased. When the concentration of GTA was 10%, the side reaction resulted in the degradation of the fibre into powder in large quantities. The optimal weight gain reached 16.4 ± 0.55% when the concentration of GTA was 2% in quaternization. Finally, the effect of reaction time on quaternization was investigated, and the results are shown in Figure 5d. With the increase in time, the fibre weight gain first increased and then decreased. The longer the reaction time was, the more adverse the quaternization reaction. The fibre weight gain reached a maximum at only 10 min, which was 16.05 ± 0.38%. Concurrently, the quaternary ammonium groups could be steadily grafted onto the fibre structure, such the optimal reaction time for quaternization was 10 min. Therefore, the optimal condition for quaternization was the immersion of 0.3 g of HF-NH₂ in 30 mL of 2% GTA ethylene glycol solution, reacted for 10 min at 70 °C.

Figure 5.

4.2. Characterisation

The FTIR spectra of HF, HF-TETA, and HF-GTA are shown in Figure 6. For the three fibres, some of the
infrared absorption peaks changed significantly. The absorption peaks between 3000 cm^{-1} and 3500 cm^{-1} for HF-
TETA and HF-GTA broadened, which was due to the superposition of the N-H in the amine or quaternary
ammonium groups with the absorption peak of the original or newly formed O-H in the hemp fibres. The
overlapping peaks near 1650 cm^{-1} corresponded to the absorption of C=N and N-H in the two modified fibres.
The absorption peak at 1102 cm^{-1} , which belonged to the stretching absorption of C-O in HF, disappeared in HF-
TETA but appeared again in HF-GTA. All these changes proved that the amino and quaternary ammonium
groups were successfully grafted onto the HF, and the reaction occurred on the hydroxyl groups of HF.

Figure 6.

The microscopic surface morphology of the hemp fibres is shown in Figure 7. The surface morphology of
the parent hemp fibre was relatively smooth. After alkalisiation and oxidation, the gullies on the surface of the
fibres significantly deepened. The surface of the fibres changed significantly and many protrusions appeared
on the surface, which were primarily caused by the grafting reaction of the amino groups in the amination
reaction. In quaternization, the protrusions became bigger, indicating that modification caused the surface of
the fibre to become rough, which could have reduced the mechanical properties of the fibre. Therefore, suitable
reaction conditions should be considered for the production of better fibres with a high density of target
functional groups and a satisfactory fibrous configuration, which could also help to increase the surface area of
the fibres and promote the process of modification. Although it was clear that GTA was grafted onto HF
throughout the four steps, from the surface morphology of fibres of each step, we found that the third
and fourth steps were very important. Improper reaction conditions led to uneven grafting and overreaction on
the fibre surface, which significantly affected the mechanical properties of the hemp fibre, which was consistent
with the real preparation procedure.

Figure 7.

Furthermore, combined with the survey scan (Figure 8) and peak table from XPS (Table 1), it can be
observed that the nitrogen content (at.%) increased sharply from HF to HF-HN₂ and gently from HF-HN₂ to
HF-GTA, which was consistent with the grafting process. The peak of chloride was also detected by XPS; these
results further confirmed that the quaternary ammonium functional groups were successfully grafted onto the
structure of the hemp fibres.

Figure 8

Table 1

XRD was used to study the crystal changes of the hemp fibres before and after modification, and the results
are shown in Figure 9. For HF, there were three diffraction peaks at approximately 14.9° (101), 16.3° ($10\bar{1}$), and
22.6° (002), which were ascribed to the cellulose I crystalline zone [23]. The diffraction peaks at 14.9° and 16.3°

became clearer, and a new weak diffraction peak appeared at approximately $34.2^{\circ}(040)$ after alkanisation.
However, only the crystalline region became smaller; the diffraction angle (2θ) of the diffraction peak remained
approximately unchanged before and after alkanisation, which indicated that the crystal structure of the fibre
was unchanged and the skeletal structure of the hemp fibre was not damaged. This was also confirmed by the
Differential Scanning Calorimeter (DSC) of HF and HF-OH in Figure 10b. From the TG curves of HF and HF-
OH in Figure 10a, it can be observed that these fibres retained similar thermal properties, with only slight changes
being observed. The degradation temperature was approximately $260\text{--}310\text{ }^{\circ}\text{C}$, which was attributed to the
thermal decomposition of hemicellulose and cellulose. This was ascribed to the equivalent crystal form of the
two fibres. For HF-CHO, HF-NH₂, and HF-GTA, the diffraction peaks were approximately equal; however,
compared with those of HF and HF-OH, the diffraction peaks changed significantly. In other words, the original
crystal form of HF was completely changed, that is, the original structure of the fibre was completely changed.
Although the latter three fibres exhibited the same crystal face from the XRD data, it can be observed from their
DSC diagrams that their melting temperature significantly varied. This meant that their structural differences
were very large, which was consistent with the experimental design. As can be observed from Figure 10a, the
thermal stability of the fibres significantly reduced due to the structural changes caused by oxidation, amination,
and quaternization, where the initial degradation temperatures were approximately $210\text{ }^{\circ}\text{C}$, $185\text{ }^{\circ}\text{C}$, and $185\text{ }^{\circ}\text{C}$,
respectively. In particular, for HF-NH₂ and HF-GTA, there were two degradation platforms at $185\text{--}330\text{ }^{\circ}\text{C}$ and
$330\text{--}380\text{ }^{\circ}\text{C}$, respectively. According to the literature [24], for hemp fibres, $260\text{--}330\text{ }^{\circ}\text{C}$ is the degradation platform
of hemicellulose, and $330\text{--}380\text{ }^{\circ}\text{C}$ is the degradation platform of cellulose. Therefore, it can be concluded that
amination and quaternization also occurred on the hemicellulose of hemp. In addition, although the pyrolysis
temperature of the hemp fibres decreased significantly after modification, it maintained good thermal stability
compared with its conventional-use environment. Furthermore, it can be observed from Fig. 10a that the five
fibres demonstrated a very similar, small weight loss platform at approximately $130\text{ }^{\circ}\text{C}$, which was caused by
the absorption of moisture and small molecular gasses in air by the fibres. This also showed that the modified
fibres retained the good moisture absorption properties of hemp fibres.

Figure 9

Figure 10

43 44 45 **4.3. Antibacterial activity against pathogens**

In this study, *E. coli* and *S. aureus* were used as the representative bacteria. Figure 11 shows that the HF and
HF-OH were not active against both pathogens, whereas the antibacterial activity sequence of the other three
fibres was as follows: HF-GTA > HF-NH₂ > HF-CHO, where the HF-GTA showed the greatest antibacterial
activity of over 95%. This was in line with the results of previous studies, which insisted that the aldehyde
groups intrinsically endowed oxidized polysaccharides with broad-spectrum antibacterial activity [25, 26]. Both
the amine and quaternary ammonium groups had positive charges, which easily interacted with the negatively-
charged bacterial phospholipid outer membrane, resulting in changes in the permeability of the bacterial
membrane, thereby causing cell membrane rupture, destruction, and dissolution [27–30]. Therefore, compared
with the amine group, the quaternary ammonium group had a stronger positive charge, such that its
antibacterial performance was stronger.

Comparatively, HF-GTA exhibited lower antibacterial activity against *E. coli* (95.41%) than against *S. aureus* (99.64%) because of the differences in the cell membrane structure between gram-negative bacteria and gram-

positive bacteria, which indicated that gram-positive bacteria, such as *S. aureus*, carried more negative charges
than gram-negative bacteria, such as *E. coli* [30, 31]. Therefore, HF-GTA was less sensitive to *E. coli*. Considering
our previous work, this novel antibacterial fibre HF-GTA showed better antibacterial activity than
ethyltriphenylphosphonium bromide-polyacrylonitrile fibre (ETPB-PANF) [6]. We speculated that the hollow
porous structure of the hemp fibres increased their antibacterial effects when combined with quaternary
ammonium groups. Coupled with the incomparable advantages of plant fibres, the hemp fibres covalently
grafted with quaternary ammonium groups are expected to become a more popular type of textile functional
fibre.

Figure 11

5. Conclusion

The hemp fibre was armed with quaternary ammonium for antibacterial activity by alkylation, oxidation,
amination, and quaternization multistage reactions. The chemical structure and micromorphology of the fibres
were characterized by FTIR spectroscopy, XPS, SEM, TGA, and XRD. Grafting and reaction mechanisms
successfully occurred, indicating that the grafting reaction primarily occurred on the hydroxyl group of
cellulose and hemicellulose in the hemp fibre, and the crystal form of the fibre changed completely after grafting.
Compared with the original hemp fibre, the thermal stability of this hemp fibre was greatly reduced, and the
initial degradation temperature was reduced to 260 °C. However, compared with the conventional application
environment, it exhibited a good thermal stability and an excellent hygroscopicity. The antibacterial activity of
three kinds of fibres containing aldehyde, amino, and quaternary ammonium groups against *E. coli* and *S. aureus*
were studied. The results exhibited that the quaternary ammonium hemp fibres gave the best antibacterial
performance, where their antibacterial ratios against *E. coli* and *S. aureus* were 95.41% and 99.64%, respectively.
The antibacterial mechanism resulted from the positive charge carried by the quaternary ammonium group
combined with the negative charge of the bacterial cell membrane, where the depletion of the electrochemical
potential on the cell membrane led to the release of cytoplasmic substances and the dissolution of cells, thereby
achieving antibacterial activity. Based on the excellent performance of HF-GTA, this work supports the
application of hemp fibres and provides a novel antibacterial material for the market.

Acknowledgments

We are particularly grateful to the Key Laboratory of Microbial Engineering, Henan Academy of Sciences Institute of Biology,
for supporting our antibacterial tests.

Funding Statement

This work was supported by the Natural Science Foundation of Hunan Province (2018JJ3584) and China Agriculture
Research System for Bast and Leaf Fibre Crops (CARS-16-E 02).

Data Accessibility

Our data are deposited at Dryad:doi.org/10.5061/dryad.z8w9ghx9m. For private access during the review period, we share our unpublished dataset using this temporary link:
<https://datadryad.org/stash/share/aVIVsydADtqc50K8HJCNadZHpCACxSmmmcDuYBCnvJA>.

Competing Interests

We declare we have no competing interests.

Authors' Contributions

L.C. participated in all the experiments, did most of the data analysis and wrote the manuscript; W.D. analyzed the data and revised the manuscript. A.C., J.L. and S.H. prepared the fibres and completed the characterization of the samples; J.T., G.P. and Y.D. completed the antibacterial experiments; L.Z. and D.L. conceived the research and provided the necessary experimental equipment. All authors gave final approval for publication.

References

[revised manuscript text omitted]

Tables

Table 1. The XPS analysis of HF, HF-NH₂ and HF-GTA

Name	HF		HF-NH ₂		HF-GTA	
	Peak B.E.(eV)	Atomic %	Peak B.E.(eV)	Atomic %	Peak B.E.(eV)	Atomic %
C1s	284.82	70.38	284.78	72.41	284.83	74.27
N1s	400.03	0.79	399.35	8.23	399.25	9.54
O1s	532.59	28.83	531.99	19.35	531.96	15.93
Cl2p					197.01	0.25

Figure and table captions

Table 1. The XPS analysis of HF, HF-NH₂ and HF-GTA.

Figure 1. Reaction mechanism for the synthesis of antibacterial hemp fibre

Figure 2. Parameters that affected the alkalization reaction. (a) Effect of NaOH concentration. At different NaOH concentrations, NaOH was mixed with 0.5 g HF in 20 mL distilled water at room temperature (20–30 °C) for 1 h. (b) Effect of soaking time. For different reaction times, 15% NaOH was mixed with 0.5 g HF in 20 mL distilled water at room temperature (20–30 °C).

Figure 3. Parameters that affect the oxidation reaction. (a) Effect of NaIO₄ concentration. At different NaIO₄ concentrations, NaIO₄ was mixed with 0.4 g HF-OH in 20 mL distilled water at 50 °C for 1 h. (b) Effect of contact time. At different reaction times, 1.5% NaIO₄ was mixed with 0.4 g HF-OH in 20 mL distilled water at 50 °C. (c) Effect of temperature. At different reaction temperatures, 1.5% NaIO₄ was mixed with 0.4 g HF-OH in 20 mL distilled water for 1 h. (d) Effect of pH value. At different pH values, 1.5% NaIO₄ was mixed with 0.4 g HF-OH in 20 mL distilled water at 50°C for 1 h.

Figure 4. Parameters that affect the amination reaction. (a) Effect of solvents. In this stage, 0.25 mL of TETA was reacted with 0.3 g of HF-CHO in different solvents at 50 °C for 6h and 500 µL of 50% glutaraldehyde was added dropwise. (b) Effect of glutaraldehyde dosage. Here, 0.25 mL of TETA was reacted with 0.3 g of HF-CHO in 40 mL acetone at 50 °C for 6 h and different contents of 50% glutaraldehyde were added dropwise. (c) Effect of contact time. At different reaction times, 0.25 mL of TETA was reacted with 0.3 g of HF-CHO in 40 mL acetone at 50 °C and 500 µL of 50% glutaraldehyde was added dropwise. (d) Effect of temperature. At different reaction temperatures, 0.25 mL of TETA was reacted with 0.3 g of HF-CHO in 40 mL acetone for 6 h and 500 µL of 50% glutaraldehyde was added dropwise.

Figure 5. Parameters that affect the quaternization reaction. (a) Effect of solvents. In this stage, 2% of GTA was reacted with 0.3 g of HF-NH₂ in different solvents at 50 °C for 10 min. (b) Effect of temperature. At different reaction temperatures, 2% of GTA was reacted with 0.3 g of HF-NH₂ in 30 mL ethylene glycol for 10 min. (c) Effect of GTA dosage. Different contents of GTA were reacted with 0.3 g of HF-NH₂ in 30 mL ethylene glycol for 10 min at 70 °C. (d) Effect of contact time. At different reaction times, 2% of GTA was reacted with 0.3 g of HF-NH₂ in 30 mL ethylene glycol at 70 °C.

Figure 6. FT-IR spectra of HF, HF-NH₂ and HF-GTA.

Figure 7. SEM images of HF (a), HF-OH (b), HF-CHO (c), HF-NH₂ (d) and HF-GTA (e).

Figure 8. XPS survey scans of HF, HF-NH₂ and HF-GTA.

Figure 9. XRD curves of HF, HF-OH, HF-CHO, HF-NH₂ and HF-GTA.

Figure 10. TG (a) and DSC (b) curves of HF, HF-OH, HF-CHO, HF-NH₂ and HF-GTA.

Figure 11. Antibacterial activity of HF, HF-OH, HF-CHO, HF-NH₂ and HF-GTA against pathogens after
disinfecting 24 h.

Figures

Figure 1. Reaction mechanism for the synthesis of antibacterial hemp fiber.

Figure 2. Parameters that affected the alkalization reaction. (a) Effect of NaOH concentration. At different NaOH concentrations, NaOH was mixed with 0.5 g HF in 20 mL distilled water at room temperature (20–30 °C) for 1 h. (b) Effect of soaking time. For different reaction times, 15% NaOH was mixed with 0.5 g HF in 20 mL distilled water at room temperature (20–30 °C).

Figure 3. Parameters that affected the oxidation reaction. (a) Effect of NaIO_4 concentration. At different NaIO_4 concentrations, NaIO_4 was mixed with 0.4 g HF-OH in 20 mL distilled water at 50 $^\circ\text{C}$ for 1 h. (b) Effect of contact time. At different reaction times, 1.5% NaIO_4 was mixed with 0.4 g HF-OH in 20 mL distilled water at 50 $^\circ\text{C}$. (c) Effect of temperature. At different reaction temperatures, 1.5% NaIO_4 was mixed with 0.4 g HF-OH in 20 mL distilled water for 1 h. (d) Effect of pH value. At different pH values, 1.5% NaIO_4 was mixed with 0.4 g HF-OH in 20 mL distilled water at 50 $^\circ\text{C}$ for 1 h.

Figure 4. Parameters that affected the amination reaction. (a) Effect of solvents. In this stage, 0.25 mL of TETA was reacted with 0.3 g of HF-CHO in different solvents at 50 $^{\circ}\text{C}$ for 6h and 500 μL of 50% glutaraldehyde was added dropwise. (b) Effect of glutaraldehyde dosage. Here, 0.25 mL of TETA was reacted with 0.3 g of HF-CHO in 40 mL acetone at 50 $^{\circ}\text{C}$ for 6 h and different contents of 50% glutaraldehyde were added dropwise. (c) Effect of contact time. At different reaction times, 0.25 mL of TETA was reacted with 0.3 g of HF-CHO in 40 mL acetone at 50 $^{\circ}\text{C}$ and 500 μL of 50% glutaraldehyde was added dropwise. (d) Effect of temperature. At different reaction temperatures, 0.25 mL of TETA was reacted with 0.3 g of HF-CHO in 40 mL acetone for 6 h and 500 μL of 50% glutaraldehyde was added dropwise.

Figure 5. Parameters that affected the quaternization reaction. (a) Effect of solvents. In this stage, 2% of GTA was reacted with 0.3 g of HF-NH₂ in different solvents at 50 °C for 10 min. (b) Effect of temperature. At different reaction temperatures, 2% of GTA was reacted with 0.3 g of HF-NH₂ in 30 mL ethylene glycol for 10 min. (c) Effect of GTA dosage. Different contents of GTA were reacted with 0.3 g of HF-NH₂ in 30 mL ethylene glycol for 10 min at 70 °C. (d) Effect of contact time. At different reaction times, 2% of GTA was reacted with 0.3 g of HF-NH₂ in 30 mL ethylene glycol at 70 °C.

Figure 6. FT-IR spectra of HF, HF-NH₂ and HF-GTA.

Figure 7. SEM images of HF (a), HF-OH (b), HF-CHO (c), HF-NH₂ (d) and HF-GTA (e).

Figure 8. XPS survey scans of HF, HF-NH₂ and HF-GTA.

Figure 9. XRD curves of HF, HF-OH, HF-CHO, HF-NH₂ and HF-GTA.

Figure 10. TG (a) and DSC (b) curves of HF, HF-OH, HF-CHO, HF-NH₂ and HF-GTA.

Figure 11. Antibacterial activity of HF, HF-OH, HF-CHO, HF-NH₂ and HF-GTA against pathogens after disinfecting 24 h.

Appendix B

Dr. Li Chang
Institute of Bast Fiber crops, Chinese
Academy of Agricultural Sciences,
Changsha, 410205, China
Phone / Fax: 086-731-88998531 (O)
Email: changli519@163.com

December 24, 2020

Editor, *Royal Society Open Science*

Dear Editor:

I wish to re-submit the manuscript titled “Improved antibacterial activity of hemp fibres by covalent grafting of quaternary ammonium groups.” The manuscript ID is RSOS-201904.

We thank you and the reviewers for your thoughtful suggestions and insights. The manuscript has benefited from these insightful suggestions. I look forward to working with you and the reviewers to move this manuscript closer to publication in the *Royal Society Open Science*.

Thank you for your consideration. I look forward to hearing from you.

The manuscript has been rechecked and the necessary changes have been made in accordance with the reviewers’ suggestions. The responses to all comments have been prepared and attached in the document called detail response to reviewers.

Thank you for your consideration. I look forward to hearing from you.

Sincerely,

Sincerely,
Li Chang

Appendix C

Dear editor and reviewers:

Thank you for your valuable advice. In light of the reviewers' comments, we have made the necessary changes to the manuscript, which have been marked in red. In addition, the data deposited in Dryad has also been updated. Point-by-point responses to the reviewers' comments are provided below.

Response to reviewers' comments

Manuscript ID: RSOS-201904

TITLE: Improved antibacterial activity of hemp fibres by covalent grafting of quaternary ammonium groups

Reviewer: 1

We sincerely appreciate your valuable comments and suggestions.

Abstract and Conclusion

1* "alkalinization", not "alkylation"

REPLY: The word "alkylation" was replaced with "alkalinisation" in the abstract and the conclusion.

Introduction

2 * P1 L2: The word "corruption" is not appropriate in this context.

REPLY: We have removed the word "corruption" and the sentence was modified to ".....cause the decomposition and deterioration of various materials".

3 * P1 L19-20: Give some data (such as FAOSTAT) related to the sentence: "To date, China remains the chief producer of hemp fibers in the world"

REPLY: The sentence "To date, China remains the chief producer of hemp fibres worldwide" was quoted from Section 1.6.2.1 of Chapter 1 in the book named "Cannabis sativa L. - Botany and Biotechnology". An excerpt from the original article is as follows: "For millennia, hemp has been a respected crop in China (Touw 1981; Clarke and Merlin 2013), where it became a very important fibre for clothing. To this day, China remains the world's chief producer of hemp fibre". After careful consideration, we the sentence should be "To date, China remains the chief producer of hemp fibres in the world according to the literature published in 2017". I tried to collect data from FAOSTAT, but failed. I'm not sure if China remains the world's chief producer of hemp fibre by 2020. So it's better to delete this sentence.

4 *P3 L37: Did the authors determine the chemical composition of raw hemp fiber?

REPLY: We did not determine the chemical composition of the raw hemp fibre. The hemp fibre we used is different from bark fibre. It is a degummed fibre that can be directly used in textiles and was provided by Hunan Huasheng Group Co., Ltd., China. According to the company, the chemical composition of degummed hemp fibre

includes cellulose (60-70%), hemicellulose (18-12%), lignin (3-7%), pectin (1-5%), lipids (0-2%), aqueous solutions (0-2%), and ash (0-3%). The difference of each component content is determined by the degumming process. The hemp fibres produced by the company can be employed in textiles, and generally require the main length to be greater than or equal to 23 mm, the metric number to be greater than or equal to 1500 Nm, and the breaking length to be greater than or equal to 26 Km.

5 * P3 L39 and P4 L6: Please use one of these options: "2,3-Epoxypropyltrimethylammonium chloride" or "Glycidyl trimethyl ammonium chloride", not both.

REPLY: The word "glycidyl trimethyl ammonium chloride" was instead of "2, 3 - Epoxypropyltrimethylammonium chloride" in the document.

6 * P3 L53: In section 3.2., there is no sufficient information regarding experimental conditions. It is better to give the alkalization, oxidation, amination and quaternization conditions. For example, 1-30% NaOH, soaking time 5-180 min, not to say: "soaked in NaOH solution for some time". I know that these conditions are presented in figures 2-5, but in my opinion, it is better to give them also in 3.2. section. Moreover, the authors should write all the solvents used in amination and quaternization reactions, not to say "different solvents".

REPLY: In Section 3.2., a detailed account of the experimental conditions has been added and marked in red.

7 * P3 L59: Do the authors determine the weight loss (%) after each treatment (i.e. alkalization, oxidation, amination and quaternization reaction)?

REPLY: The weight gain after amination and quaternisation had been provided in the manuscript. We have also added the weight loss data of the alkalisation and oxidation reactions in the manuscript (Figure 2 and 3), along with the corresponding analyses.

Figure 2. Parameters that affect the alkalisation reaction. (a) Effect of NaOH concentration. At different NaOH concentrations, 0.5 g HF was immersed in NaOH for 60 min at room temperature (20–30 °C). (b) Effect of soaking time. For different reaction times, 0.5 g HF was immersed in 15% NaOH at room temperature

(20–30 °C).

Figure 3. Parameters that affect the oxidation reaction. (a) Effect of NaIO₄ concentration. At different NaIO₄ concentrations, 0.4 g HF-OH was immersed in NaIO₄ for 60 min at 50 °C. (b) Effect of contact time. For different reaction times, 0.4 g HF-OH was immersed in 1.5% NaIO₄ at 50 °C. (c) Effect of temperature. At different reaction temperatures, 0.4 g HF-OH was immersed in 1.5% NaIO₄ for 60 min. (d) Effect of pH value. At different pH values, 0.4 g HF-OH was immersed in 1.5% NaIO₄ for 60 min at 50 °C.

8 * P4 L: How do you determine the content of aldehyde groups? Describe the method in section 3.3.

REPLY: The method for determining the content of aldehyde groups has been added in Section 2.2.

Results and Discussion

9 * Through all manuscript and figures: "aldehyde group content", not "aldehyde content"

REPLY: The term "aldehyde content" was replaced with the phrase "aldehyde group content" throughout the manuscript and in all the relevant figures.

10 * P5 L14-L15: Can you describe the sentence: "To release more reactive hydroxyl groups, NaOH was used during alkanilization process". The hydroxyl groups were not released after the alkanilization, their availability was increased. Please establish the connection between the NaOH treatment and hemp non-cellulosic components. See the already cited reference 12 (Kostic et al. 2010) and the newest published literature

from the same author such as:

Ivanovska, A., Asanovic, K., Jankoska, M. et al. Multifunctional jute fabrics obtained by different chemical modifications. *Cellulose* 27, 8485-8502 (2020). <https://doi.org/10.1007/s10570-020-03360-x>

REPLY: The sentence: "To release more reactive hydroxyl groups, NaOH was used during alkanilisation process" has been corrected to "To produce more reactive hydroxyl groups, ...".

In Reference 12, the purpose was to study the influence of hemicellulose and lignin removal on the adsorption and electric properties, as well as the fineness of hemp fibres. The fibres they studied were bark fibres. An excerpt from the results and discussion section of the paper is as follows:

"Surface changes of hemp fibres as a result of the separate removal of hemicelluloses and lignin can be seen by comparison of scanning electron micrographs of unmodified and modified hemp fibres presented in Fig. 2. As a natural bast fibre, hemp showed great variation in multi-cellular fibre diameter. The fibre surface appeared relatively rough and uneven, the fibril bundles within the fibre seemed to be embedded in a somewhat soft".

Instead of bast fibres, however, we have used degummed hemp fibres that can be directly used in textiles. The purpose of degumming is to remove pectin, lignin and other impurities from raw hemp fibres to obtain soft and loose fibres. Following this, bleaching and other processes finally result in hemp fibres suitable for textile use. However, the fibres used in this study already had good textile properties, thus eliminating the need for further processing. The objective herein is to impart the fibres with antibacterial properties through chemical modification. In the study, it was found that fibres treated without sodium hydroxide reacted directly with sodium periodate and a small amount of aldehyde groups was generated. However, the higher the content of the aldehyde groups, the more the grafting rate of amine groups and quaternary ammonium groups. In other words, the higher the grafting rate, the better the antibacterial activity. Therefore, "To produce more reactive hydroxyl groups, NaOH was used during the alkanilisation process".

The paper "Multifunctional jute fabrics obtained by different chemical modifications (2020)" was the first to reduce the jute fabric non-cellulosic components by using different chemical modifications (i.e. alkali and oxidative) and analyse their influence on the jute fabric properties.

Hemp and jute fibres are used for different applications. Hemp fibre is softer and has better properties that can be used for manufacturing clothing, while jute is coarser and is mainly used for making ropes, sacks, paper, carpets and curtains. Jute fibre is not suitable for clothes, owing to its itchy nature. This may have something to do with their chemical composition and fibre structure.

We could try to establish the connection between the NaOH treatment and hemp non-cellulosic components in future studies. However, this will not affect our access to antibacterial hemp fibres. Anyway, thank you very much for your advice, we will consider it seriously.

* P5 L13-34: The discussion in this section is unclear. Try to better describe the results presented by using the following literature: D. Klemm, B. Philipp, T. Heinze, U. Heinze, W. Wagenknecht, "Comprehensive Cellulose Chemistry", Volume 1, Fundamentals and Analytical Methods, Wiley-VCH, Weinheim, 1998. (pp. 56 and 99).

REPLY: We have modifications to this section for better clarity.

* The content of aldehyde groups in raw hemp fibers should be given in Figs. 2 and 3.

REPLY: Raw hemp fibres do not contain aldehyde groups, and no aldehyde groups were generated after reaction with NaOH solution. The alkalisated fibre, HF-OH, was reacted with NaIO₄ solution, subsequently, C2-OH and C3-OH of glucose broke down to form aldehyde groups. Figure 2 shows the optimal NaOH concentration and alkanisation time determined according to the content of aldehyde groups formed by the hydroxyl groups after the oxidation reaction. Figure 3 displays the optimal conditions for the oxidation reaction of raw fibres after being treated with 15% NaOH for 60 min. Therefore, the following sentence has been added in Section 3.1.1 as follows: " Notably, aldehyde groups are absent in raw hemp fibres. The optimal NaOH concentration ..."

*P5 L21: "up to 15%", not "below 15%"

REPLY: "below 15%" was changed to "up to 15%".

* P5 L24-25: Support the statement "degradation of cellulose by the high-concentration alkali" by giving the results for cellulose degree of polymerization or some literature data.

REPLY: After consulting many literatures, it is found that the alkali degradation reaction of cellulose occurs under high temperature conditions or in the presence of urea and Li⁺ as catalysts. The alkanisation reaction in the manuscript occurred at room temperature to swell the fibres and increase the accessibility of hydroxyl groups. The cellulose did not degrade. Therefore, our statement that "degradation of cellulose by the high-concentration alkali" was incorrect. The correct statement should be that " When the concentration was further increased, the aldehyde group content significantly decreased, which was ascribed to the fact that the high concentration of alkali increased the metal ions in the solution to a certain extent, whereas the radius of hydrated ions decreased due to the high concentration of ions, resulting in a decrease in swelling degree and the effect of alkanisation". We have also made modifications to the manuscript accordingly.

* P5 L31: During the treatment with NaOH the hydroxyl groups were not separated, their availability increased due to the removal of non-cellulosic components such as hemicelluloses, fats, waxes and pectin.

REPLY: Yes, you are absolutely right. The sentence "Therefore, it is critical to first treat hemp fibres with NaOH to separate more active hydroxyl groups". was corrected to

"Therefore, it is critical to first treat hemp fibres with NaOH to produce the reactive hydroxyl groups".

16 * Description of Fig. 2b. Did the authors want to say: 20 ml 15% NaOH? or they further dilute 15% NaOH by adding 20 ml water? Make appropriate corrections in Fig. 3.

REPLY: The sentence in Figure 2 was corrected to "(a) Effect of NaOH concentration. At different NaOH concentrations, 0.5 g HF was immersed in NaOH for 60 min at room temperature (20–30 °C). (b) Effect of soaking time. For different reaction times, 0.5 g HF was immersed in 15% NaOH at room temperature (20–30 °C). "

The sentence in Figure 3 was corrected to "(a) Effect of NaIO₄ concentration. At different NaIO₄ concentrations, 0.4 g HF-OH was immersed in NaIO₄ for 60 min at 50 °C. (b) Effect of contact time. For different reaction times, 0.4 g HF-OH was immersed in 1.5% NaIO₄ at 50 °C. (c) Effect of temperature. At different reaction temperatures, 0.4 g HF-OH was immersed in 1.5% NaIO₄ for 60 min. (d) Effect of pH value. At different pH values, 0.4 g HF-OH was immersed in 1.5% NaIO₄ for 60 min at 50 °C."

17 * P5 L49-53: The authors should describe how the NaIO₄ concentration affected the aldehyde group content.

REPLY: The NaIO₄ concentration affected the aldehyde group content has been described as follows: "The aldehyde group content increased with increasing NaIO₄ concentration; a sharp increase (244 ± 19 to 541 ± 17 μmol/g) from 0.5% to 1.5% could be observed; the highest increase was up to 541 ± 14 μmol/g at 1.5% or 2.0% NaIO₄. When the concentration of NaIO₄ was more than 2.0%, there was a lower aldehyde group content. Therefore, the proper concentration of NaIO₄ was determined as 1.5% to maintain a good fibre shape and low cost."

18 * P7 L5: "providing that the temperature did not have significant effect...", not "providing that contact time did not have significant effect..."

REPLY: The sentence was corrected to "proving that the temperature did not have significant effect..."

19 * P8 L1-10: Omit the usage of the word "absorption" in the case of description of the FTIR transmittance spectra.

REPLY: The word "absorption" in FTIR transmittance spectra was omitted.

20 * Please give the FTIR spectra of HF-OH and HF-CHO and describe the changes occurring due to the treatment with NaOH and NaIO₄.

REPLY: The FTIR spectra of all five fibres are given in Figure 6, and the corresponding analyses have been added in the manuscript. The description is as follows: "The FTIR spectra of HF, HF-OH, HF-CHO, HF-NH₂, and HF-GTA are shown in Figure 6. The peaks at 3287 and 3332 cm⁻¹ are ascribed to the stretching vibration of hydroxyl groups with inter-molecular hydrogen bonds and intra-molecular hydrogen bonds,

respectively. After alkylation, the peaks at these positions become wider, and two small peaks appear at 3439 and 3485 cm^{-1} , which indicates that the hydrogen bonds in hemp molecules are destroyed by alkylation, and the bound water is introduced into the reaction process. The peak at 1638 cm^{-1} is due to the stretching vibration of C=C, N-H, or C=N, for HF, HF-OH and HF-CHO, the peak is mainly caused by the stretching vibration of C=C on the benzene ring of lignin, and the peak at 1104 cm^{-1} is also mainly produced by lignin, and both of them are weakened after alkylation, which proves that a part of lignin is dissolved during alkylation reaction. After amination, the peak is significantly enhanced, which is due to the superposition of C=N stretching vibration, N-H deformation vibration, and C=C stretching vibration. The peak was slightly weakened after quaternisation, which also proved that the quaternisation reaction occurred on N-H. HF-CHO has a weak peak at 1717 cm^{-1} , which is caused by the stretching vibration of aldehyde group formed by oxidation. The asymmetric and symmetric stretching vibrations of the C-O bonds (C-O-C and C-OH) and the substituted phenyl C-H can be observed in the region of 1185 to 837 cm^{-1} (1160, 1105, 1053, 1029, 1001, 985, and 897 cm^{-1}). The infrared spectra of the five fibres proved that the chemical grafting reaction was successfully completed, and the reaction mainly occurred on the hydroxyl groups of HF".

Figure 6. FT-IR spectra of HF, HF-OH, HF-CHO, HF-NH₂ and HF-GTA.

21 * P8 L32: "third and fourth steps", not "thirdandfourthsteps"

REPLY: The mistake "thirdandfourthsteps" was corrected to "third and fourth steps".

22 * From Fig. 8 it is evident that the highest content of nitrogen was detected in HF sample, while the lowest in HF-GTA sample. The presented results do not match with the discussion. Also, the chloride peak was not detected in the samples.

REPLY: We are very sorry that the order of the fibres HF, HF-HN₂ and HF-GTA in Figure 8 was marked incorrectly. The correct figure is shown below. According to XPS results, chloride did not exist on HF and HF-NH₂ while 0.25 at% of it was present on HF- GTA.

Figure 8. XPS survey scans of HF, HF-NH₂ and HF-GTA.

23-24 * P8 L56- P9 34: Please use Cellulose 21:885-896 (2014) for the required nomenclature for X-ray diffraction peak labels and accordingly correct discussion in this section.

* After the treatment with 15% NaOH, the structure of cellulose was changed, part of the cellulose I was converted to cellulose II. Please use Cellulose 21:885-896 (2014) for the required nomenclature for X-ray diffraction peak labels (for cellulose I and cellulose II) and accordingly correct discussion in this section.

REPLY: We are very sorry that we have given some inaccurate discussions on XRD patterns in the manuscript. We have carefully read the references recommended by you and revised the manuscript. At the same time, we found that our data graphs are different from the references. Therefore, we re-obtained the XRD patterns of the HF and HF-OH samples. The results show that due to the carelessness of the testing company, the order of the HF and HF-OH samples was wrong. Therefore, we adjusted the labels in the XRD and have explained them carefully. The detailed modified content is as follows: "XRD was employed to study the changes in the crystal structure of the hemp fibres before and after modification, and the results are shown in Figure 9. For HF, there were three diffraction peaks at approximately 14.9° (1-10), 16.3° (110), and 22.6° (200), which were ascribed to the cellulose I_β crystalline zone [25,26]. The diffraction peaks at 16.3° became weaker. Simultaneously, there grew a weak shoulder peak on the diffraction peak of 22.6°, which was caused by the transformation of cellulose I_β to cellulose II during the alkalisation. However, the conversion was not extensive. This phenomenon was also reported by Prof. Dr. Klemm D et al. [27], who observed that phase transition occurs within the regions of the crystalline order when lye concentrations exceeded 12-15%. This transformation is attributed to the intracrystalline swelling caused by the inclusion of NaOH and H₂O into the crystallites. This was also confirmed by the differential scanning calorimetry (DSC) analysis of HF and HF-OH (Figure 10b), in which the peak temperature of the melting endothermic peak is nearly the same. In addition, the crystallinities of HF and HF-OH fibres are calculated using formula (4) [28] to be 75.63% and 73.97%, respectively, implying that the crystallinity of HF-OH fibres decreased. However, the

crystal structure of the fibre is completely altered by the oxidation reaction, and the crystal structure conforms to the cellulose II crystalline zone. Moreover, the three main peaks at 11.82°, 19.92° and 21.61° corresponded to the Miller indices of (1-10), (110) and (020). The crystallinity of HF-CHO decreased to 67.17%. The crystal structure of HF-NH₂ and HF-GTA was similar to that of HF-CHO, but the crystallinity decreased to 60.48% and 60.35% after amination and quaternisation."

Figure 9. XRD curves of HF, HF-OH, HF-CHO, HF-NH₂ and HF-GTA.

25 * The authors stated that after the alkali treatment, the "crystalline region became smaller". How did the authors determine the fibers' crystallinity?

REPLY: The crystallinities of the HF and HF-OH fibres were calculated using Formula (4) to be 75.63% and 73.97%, respectively. We have added the relevant details about the crystallinity calculation in the manuscript.

$$I_{Cr} = \frac{I_{200} - I_{am}}{I_{200}} \times 100\% \quad (4)$$

where I_{Cr} is the crystallisation index, I_{200} is the diffraction peak intensity of the crystal plane (200), and I_{am} is the intensity of the diffraction peak in the amorphous region ($2\theta = 19.0^\circ$ for cellulose I).

26 * Fig. 9. The changes in the XRD diffraction patterns were not clearly described.

REPLY: We have clearly explained the changes in XRD in the manuscript.

27 * Why the authors presented the TG and DSC analysis? In my opinion, these methods are not essential for characterizing the antimicrobial hemp fibers.

REPLY: Elucidating the thermal properties of the fibres can prove useful. Good thermal stability is very important for the subsequent processing of the fibre and its products (such as heat setting, dyeing, washing and finishing). For crystalline fibres such as cellulose, the DSC and XRD data can be used to analyse the crystallisation characteristics of the fibres. Moreover, the crystallinity and its change can be inferred

by the melting enthalpy.

28 * Having in mind four successive chemical reactions, the authors should determine the hemp fibers' mechanical properties. If the treated hemp fibers have not good mechanical properties, what will be their end-use?

REPLY: Chemical reactions often damage or even destroy the structure of the fibre, resulting in a decline of its mechanical properties and even pulverisation. In this study, the multi-step chemical modification also has a detrimental impact on the mechanical properties of the fibre. However, through the optimisation of the modification process, the degree of grafting and fibre morphology were balanced. Therefore, although the strength of the prepared fibre decreased to a certain extent, it still maintained a good fibre morphology and a certain mechanical strength. In real-life applications, through fibre blending and other means, it can fully meet the requirements of use.

29 * In many cases, washing durability is a key parameter for assessing the performance of antimicrobial fibers. How much of the antimicrobial effect survives after repeated washings is the major issue addressed by most researchers. The authors should test the antimicrobial durability of chemically treated hemp fibers?

REPLY: Discussion about the washing durability of the hemp fibres has now been in the Result section 3.4 in this manuscript. The HF-GTA fibres were washed multiple times using a 0.1% neutral hand detergent for 20 min, then rinsed with distilled water and dried overnight at 40 °C. A fibre sample (0.2 g) was subjected to antibacterial tests post treatment. It is evident from the results shown in Table 2 that the antibacterial ratios of HF-GTA exposed to *E. coli* and *S. aureus* for 24 h decreased with the increasing number of wash cycles. However, the antibacterial activity was retained even after 30 washes. The antibacterial ratios of HF-GTA against *E. coli* and *S. aureus* remained 89.78% and 91.12%, respectively, indicating that HF-GTA is endowed with an appreciable washing resistance. Therefore, HF-GTA can be employed in the development of various antibacterial functional textiles and the sterilisation of industrial and civil water.

Table 2. Washability of HF-GTA

Number of washes	Antibacterial ratio (%)	
	E. coli	S. aureus
0	95.41	99.64
10	93.31	96.67
30	89.78	91.12

30 * Abstract and Conclusion:

Rewrite the abstract according to the given suggestions in the section "Results and discussion".

REPLY: We have carefully revised and improved the abstract and conclusion.

Reviewer: 2

Comments to the Author(s)

The results are interesting since new antibacterial materials need to be investigated for the actual pandemic, but it needs a major revision. Check attached file!

Carefully comparing the main document, the comments of the reviewer 2 are as follows:

We are deeply appreciative of your valuable comments and the editor's hard work in sorting out them!

We have been polished the language for a clear and concise presentation, and the detailed modifications are marked in red.

The author claims have obtained long-lasting antibacterial fibres for textile uses, but, quaternary ammonium salts degrades if exposed to the light, so, another applications should be suggested.

REPLY: Quaternary ammonium salt is indeed unstable under special conditions, especially under strong ultraviolet light. However, it is only relatively unstable. Overall, its stability is decent, and this relative instability does not hinder the development and application of quaternary ammonium salt products. Quaternary ammonium salts antibacterial finishing agents, especially organosilicon quaternary ammonium salts finishing agents, have been widely used in the textile, leather, metal, coating, and daily chemical industries. In addition, quaternary ammonium salts are also the main functional groups of strong-basic anion exchange resin. For a long time, the application of strong-basic anion exchange resin products has been widely accepted, and they are regarded as standard products. This proves that there is no alarming concern toward the practical application of quaternary ammonium salts. The quaternised hemp fibre prepared in this study can be used in textiles, especially in the preparation of skin-contact antibacterial textiles (such as socks, underwear, towel, bath towel, mattress, etc.), and can also replace strong basic ion exchange resin.

space between "furthermore-hemp"

REPLY: "furthermore,hemp" was corrected to "furthermore, hemp".

It should be in plural "Quaternary ammonium salts are widely used as efficient disinfectants..." Even in the next line

REPLY: The term "Quaternary ammonium salt" was corrected to "Quaternary ammonium salts" in plural.

must be "salts"

REPLY: The word "salt" was corrected to "salts".

have been very few instead of "have been rare"

REPLY: The sentence was changed to "studies based on the modification of hemp fibres for antibacterial properties have been rarely reported."

count means diameter?

REPLY: I am sorry for my unprofessionalism. "Count" should be "metric number". It does not mean diameter. It is a unit used to indicate the thickness of a textile fibre. The higher the metric number, the finer the fibre.

"until weight stopped varying" instead of "to obtain a constant weight"

REPLY: The weight of the fibre was altered through the chemical modification. After the unreacted reagent on the surface of the fibre is cleaned with distilled water, it is put into the oven, which is used to dry the fibre at 40 °C. The drying standard is that the fibre weight remains unchanged after water removal. Therefore, the sentence was corrected to "to constant weight" is probably more appropriate.

room temperature instead of certain

REPLY: The specific experimental temperature has been added in section 2.2.

until weight stopped varying

REPLY: Similar to the response 7, the sentence was corrected to "to constant weight".

10-11 the solvents, temperature and the time MUST be specified!

again, solvents, temperatures and time of the experiments MUST be specified!

REPLY: The specific experimental conditions have been added in section 2.2.

In figure 1, the purpose of adding NaOH is deprotonate, so, Why didn't you show it in the scheme?

REPLY: We have modified the reaction scheme, and added hydrogen bonds and alkalized groups in the Scheme 1.

eliminate the a in "a 1 mL"

REPLY: The extra word "a" was deleted.

the sapace between C and was must be added

REPLY: The space has been added, and the sentence was changed to "...a temperature of 40 °C...".

This paragraph must be in the conclusions

REPLY: The paragraph has been modified as " So the optimal oxidation conditions were pH 4.83, 1.5% NaIO₄, heated to 40 °C for 60 min; this coincided with the reaction mechanism. During this reaction, NaIO₄ had a strong oxidation effect, and its oxidation became stronger with increasing concentration, temperature, and time. However, the aldol condensation reaction occurred because the aldehyde group formed by over oxidation of the hemp fibres and the hydroxyl group of cellulose, such that the hemp fibres were severely dissolved and became stiff. Therefore, controlling the degree of oxidation reaction provided a good matrix material for the next step. " In addition, the conclusion has been modified accordingly.

the absorption signal at 1650 cm^{-1} appears in the three samples, but why it appears in sample 1 if it does not have an C=N or N-H groups?

REPLY: The FTIR spectra of all samples were recharacterised with a Bruker Invenio spectrometer (Invenio, Germany) with a Platinum ATR module (equipped with a diamond crystal). The wavenumber range was 400 to 4000 cm^{-1} , and the resolution was 4 cm^{-1} . Each sample was scanned 16 times. The peak at 1650 cm^{-1} in the original manuscript corresponds to 1638 cm^{-1} in the new spectra. Moreover, the peak at 1638 cm^{-1} is ascribed to the stretching vibration of C=C, N-H or C=N. For HF, HF-OH, and HF-CHO, the peak is mainly induced by the stretching vibration of C=C on the benzene ring of lignin.

The antibacterial mechanism resulted from the positive charge carried by the quaternary ammonium group combined with the negative charge of the bacterial cell membrane, why do you conclude that if the GTA has a positive and negative charge?

REPLY: The anion corresponding to the cationic quaternary ammonium groups in the synthetic quaternary ammonium hemp fibre is chloride ion. However, all the antibacterial experimental systems in this study contain 0.9% sodium chloride solution, and the concentration of negative chloride ion is very high. Under this condition, good antibacterial performance can be obtained. It is evident that the presence of negatively charged chloride ions does not affect the antibacterial performance of the fibres.

Appendix D

Dear editor and reviewers:

Thank you for your valuable advice. In light of the reviewers' comments, we have made the necessary changes to the manuscript, which have been marked in red. Point-by-point responses to the reviewers' comments are provided below.

Response to reviewers' comments

Manuscript ID: RSOS-201904

TITLE: Improved antibacterial activity of hemp fibres by covalent grafting of quaternary ammonium groups

Reviewer: 1

Comments to the Author(s)

The quality of the manuscript is improved, however, some additional corrections should be made.

We sincerely appreciate your valuable comments and suggestions.

P3 L2: Please insert the chemical composition of hemp fibers obtained by the company.

REPLY: Dear reviewer, I am very sorry for the mistake that the data provided last time is the chemical composition of hemp bast fibres without degumming. We have rechecked with the company and included the correct data in the manuscript "Degummed hemp fibres (cellulose > 90%, hemicellulose 2-3%, lignin < 1%, pectin < 1%, lipids < 1%) with a length of approximately 5 cm and the metric number of 1568 Nm were obtained from Hunan Huasheng Group Co., Ltd., China."

P3 L17: As I can see in Fig. 3d, the pH range is from 3.5 to 11, not from 3 to 11.

REPLY: The pH range was modified to from 3.5 to 11.

P3: Rewrite the method for the determination of aldehyde group content. For example: "3-5 drops of thyme blue indicator were added to 50 mL of 20 g/L

hydroxylamine hydrochloride methanol solution”, not “take 50 mL of 20 g/L hydroxylamine hydrochloride methanol solution and add 3-5 drops of thyme blue as indicator.” Moreover, check the indicator name, I am not sure that it is thyme blue. Give some reference where the mentioned procedure was used.

REPLY: We have provided the references in the manuscript. The method was modified as "The content of aldehyde groups was determined using the method of hydroxylamine hydrochloride [22,23] with minor modifications as follows: 3-5 drops of thymol blue indicator were added to 50 mL of 20 g/L hydroxylamine hydrochloride methanol solution. If the solution is red, 0.03 mol/L sodium hydroxide methanol solution was added dropwise until the solution becomes yellow;". And the indicator name has been corrected to thymol blue.

[22] Maute RL, Owens ML. 1956 Rapid Determination of Carbonyl Content in Acrylonitrile. *Anal. Chem.* 28, 1312-1314. (doi: 10.1021/ac60116a024)

[23] Hou QX, Liu W, Liu ZH, Bai LL. 2007 Characteristics of Wood Cellulose Fibers Treated with Periodate and Bisulfite. *Ind. Eng. Chem. Res.* 46, 7830-7837. (doi: 10.1021/ie0704750)

The captions of figures 2, 3, 4 and 5 are so long, rewrite them.

REPLY: The captions of figures have been simplified as:

"Figure 2. Effects of NaOH concentration (a) and soaking time (b) on alkalisation reaction.

Figure 3. Effects of NaIO₄ concentration (a), contact time (b), temperature (c) and pH value (d) on oxidation reaction.

Figure 4. Effects of solvents (a), glutaraldehyde dosage (b), contact time (c) and temperature (d) on amination reaction.

Figure 5. Effect of solvents (a), temperature (b), GTA dosage (c) and contact time (d) on quaternisation reaction."

Moreover, the parameters were changed to the corresponding results and discussion instead of the caption.

Fig. 2 and 3: "aldehyde group content", not "aldehyde content".

REPLY: The ordinates in Figure 2 and Figure 3 have been changed to "Aldehyde group content".

P4 L26, L54: The authors did not understand my comment. The hydroxyl groups were not released or produced, their availability increased after the treatment with sodium hydroxide. Namely, hydroxyl groups are already present in the fibers, and as a result of sodium hydroxide treatment (which selectively removed hemicelluloses), their availability increased.

REPLY: The sentence was corrected to "To increase the availability of hydroxyl groups, NaOH was used during the alkalisation process,".

P4 L32-38: The authors should give some reference/s.?

REPLY: The references have been provided in the manuscript, and the details of the references are as follows:

[25] Zhang HJ, Zhong ZL, Wang YX, Liao ZD. 2014 Effect of activation time on the structural performance of hemp (*In Chinese*). *J. Cellulose Sci. Technol.* 22, 39-43. (doi: 10.3969/j.issn.1004-8405.2014.04.007)

[26] David NSH. 1996 Chemical modification of lignocellulosic materials. New York: Marcel Dekker, Inc. 48-51.

P4 L38: This sentence is incorrect: "Notably, aldehyde groups are absent in raw hemp fibres." The aldehyde groups are present in the lignin. Since you have about 3-7% lignin, I am sure that the raw hemp fibers have a small amount of aldehyde groups. Please determine it and present the results in the figures. The small amount of aldehyde groups were also found in other bast fibers such as flax and jute. Check the following literature:

Milanovic et al. 2012 Influence of TEMPO-Mediated Oxidation on Properties of Hemp Fibers

Ivanovska et al. 2020 Waste Jute Fabric as a Biosorbent for Heavy Metal Ions from Aqueous Solution

Lazic et al. 2017 Influence of hemicelluloses and lignin content on structure and sorption properties of flax fibers (*Linum usitatissimum* L.)

REPLY: The hemp fibre we employed in the study had been pre-treated by degumming and other processes by the provider, and it is very loose and soft. In fact, it should be called degummed hemp fibre, and its photo is shown in the figure below. So, the alkalisation process described in the study can also be called secondary degumming and activation. It is reported in the literature that the fibrils such as jute, flax and hemp contain many other components (hemicellulose, Lignin, Pectin, etc.). Among them, hemicellulose contains a certain amount of aldehyde groups. Since the alkali treatment can easily remove the hemicellulose, the aldehyde group content of

the fibre after alkalisation also will decrease. For example, when the flax fibre is treated with 18% NaOH solution for 60 minutes, the aldehyde group content is reduced from 0.22 to 0.008 mmol/g, but the flax fibre after the treatment still contains 3.44% hemicellulose. (Lazić, B.D., Pejić, B.M., Kramar, A.D. et al. Influence of hemicelluloses and lignin content on structure and sorption properties of flax fibres (*Linum usitatissimum* L.). *Cellulose* 25, 697–709 (2018). <https://doi.org/10.1007/s10570-017-1575-4>). This may be due to the complex composition of hemicellulose, most of the hemicellulose containing aldehyde groups is removed by alkalisation, and the content of aldehyde groups in the remaining hemicellulose is very low. And we measured the aldehyde group content of the original hemp fibre again, but the aldehyde group was still not detected.

Fig. Degummed hemp fibre can be directly used for textile

P4 L58: The weight loss obtained after the sodium hydroxide treatment is very high (between 15 and 30%). Why? Which are the coexisting components in the hemp fibers that are removed during the alkalization?

REPLY: The hemp fibre we employed in the study had been pre-treated by degumming and other processes by the provider. In fact, it should be called degummed hemp fibre. So, the alkalisation process described in the study can also be called secondary degumming and activation. After the hemp fibre undergoes alkalisation, the fibre bundles are "unbundled" under the action of the lye. Some substances such as pectin that bind the fibres will be dissolved in the lye, and the fibre bundles will spread out. Correspondingly, the more obvious this "unbundling" phenomenon is, the weaker the strength of the fibre will be. For shorter and thinner fibre filaments, in addition to the dissolution of pectin and other substances in the fibres, it will also cause the decomposition of the fibre itself, and disperse into the lye. So the weight loss rate of the fibre we measured is higher.

P5 Rewrite section 3.1.2. The authors describe all that we can see in the figures. However, they do not conclude why such results are obtained. Correlate

experimental conditions with weight loss and aldehyde group content.

REPLY: We have rewritten this section and marked it in red in the manuscript.

Modification: The conversion of hydroxyl groups on hemp fibre surface to aldehyde groups can provide chemical reaction sites for Schiff base reaction of amination of hemp fibre. So far, the selective oxidation system for the secondary hydroxyl group is only the periodate, which was discovered in 1937 [28]. Therefore, we choose NaIO_4 as the oxidant. Furthermore, the effect of the concentration of NaIO_4 , oxidation time, temperature, and pH on the aldehyde group content of HF-CHO was investigated. The HF-OH (0.4g) was reacted with different concentrations of NaIO_4 at 50 °C in the dark at 180 rpm for 60 min. Figure 3a shows that aldehyde group content increased with increasing NaIO_4 concentration. The maximum content was $541 \pm 14 \mu\text{mol/g}$ at 1.5% or 2.0% NaIO_4 with good fibrous morphology. When the concentration of NaIO_4 was over 2.0%, the aldehyde group content decreased, which may be attributed to the fact that periodate gradually penetrates, diffuses and reacts from the amorphous region to the crystalline region of the hemp fibre during the reaction [29]. The macromolecules in the amorphous region are poorly arranged and the structure is loose, which is conducive to the penetration and reaction of reagents. However, the macromolecules in the crystalline area are arranged neatly and densely, and the gap space in the fibre is small, which is not conducive to the penetration and diffusion reaction of the reagent, and the accessibility of the reagent to the fibre in the region is poor. On the contrary, in the presence of a high concentration of oxidants, the aldehyde groups generated by oxidation will undergo an aldol condensation reaction with unreacted hydroxyl groups on the cellulose, resulting in a slight decrease in the content of aldehyde groups [30]. The concentration of NaIO_4 does not have a significant effect on the weight loss of fibre Within the scope of the investigated conditions, but when the concentration of NaIO_4 exceeds 2%, the surface of fibre will become stiff and lose the original fibrous morphology. Therefore, the proper concentration of NaIO_4 was determined as 1.5% to maintain a good fibre shape and low cost. For different reaction times, 0.4 g HF-OH was reacted with 1.5% NaIO_4 at 50 °C in the dark at 180 rpm. Figure 3b shows that there was a sharp increase in the aldehyde group content from 20 min to 60 min, and the maximum content of the aldehyde group was $541 \pm 3.06 \mu\text{mol/g}$ at 60 min. Thereafter, the aldehyde group content slowly decreased until 120 min. With the prolongation of oxidation time, the degree of oxidation increased, and the content of aldehyde group increased continuously. However, longer oxidation time (> 60min) tends to condensation reaction between the generated aldehyde group and the hydroxyl group of cellulose, resulting in the decrease of the content of aldehyde group. The reaction time has no significant effect on the weight loss of fibre within the scope of the investigated conditions, but the longer the reaction time, the stiffer the fibres. Therefore, a reaction time of 60 min was optimal. At different reaction temperatures,

0.4 g HF-OH was reacted with 1.5% NaIO₄ in the dark at 180 rpm for 60 min. Figure 3c shows that a higher temperature enhanced the aldehyde group content of the final fibres. However, when the reaction temperature was 70 °C, it not only promotes the occurrence of the comprehensive aldol reaction, but also enhances the penetration of the oxidant into the crystal area, resulting in a decrease in the content of aldehyde groups, while part of the crystal area of the fibre is also destroyed, thereby the weight loss of the fibre also was increased. Hemp fibre obtained the highest aldehyde group content at 60°C, but the fibre morphology was not as good as that at low temperature because of its high oxidation degree. Therefore, a temperature of 40 °C was considered as the most favourable reaction temperature. At different pH values, 0.4 g HF-OH was reacted with 1.5% NaIO₄ in the dark at 180 rpm for 60 min. Figure 3d shows that at a pH of 4.83, which was the pH of the original NaIO₄ solution, the maximum content of aldehyde group (543 ± 3.78 μmol/g) was received; other pH values were not conducive to the progress of oxidation, especially in the alkaline environment. Under alkaline conditions, the hydrated sodium ions in the solution will enter the crystalline area of the fibre, increasing its swelling degree [6], so that periodate is more likely to enter the crystalline area and destroy it, and ultimately cause a serious loss of fibre strength. That is, the weight loss of the fibre in Figure 3(d) increases significantly with pH above 5. Of course, the loss of fibre structure must also be accompanied by a significant decrease in the content of aldehyde groups. Therefore, a pH of 4.83 (i.e., NaIO₄ directly dissolved in water) was a suitable value during the synthesis of HF-CHO. In short, NaIO₄ had a strong oxidation effect, and its oxidation became stronger with increasing concentration, temperature, and time. However, the aldol condensation reaction occurred because the aldehyde group formed by over oxidation of the hemp fibres and the hydroxyl group of cellulose, such that the hemp fibres were severely dissolved and became stiff. Therefore, controlling the degree of oxidation reaction provided a good matrix material for the next step. So, the optimal oxidation conditions were pH 4.83, 1.5% NaIO₄, heated to 40 °C for 60 min; this coincided with the reaction mechanism.

Section 3.2. Try to better describe the FTIR spectra of the samples. I can not fully agree with the authors. Namely, after the treatment with NaOH, the band between 3600 and 3000 cm⁻¹ was broadened due to the increased availability of cellulose hydroxyl groups (caused by the removal of non-cellulosic components, which was earlier proved by the weight loss after the alkalization). Additionally, after the sodium periodate treatment, the intensity of this band decreased since the hydroxyl groups were converted to aldehyde groups... Also, the authors did not give any reference/s in this section.

REPLY: After careful comparison of the data and literature, although the hemp fibre

used in this study is degummed hemp fibre, it still contains some non-cellulose components. So, the change in the IR spectra of the hydroxyl groups after alkalinisation cannot be attributed only to the destroy of hydrogen bond, but should include the dissolution of some non-cellulose substances. After the literature investigation, the decrease of the intensity of hydroxyl peak in the oxidized fibre can be used to infer the conversion of hydroxyl to aldehyde group. Therefore, we have revised the relevant parts with reference to your suggestions, and provided the references for this section. And all the changes are marked in red in the manuscript.

The authors choose to use “alkalinisation” instead of “alkalinization” throughout all manuscript. According to that, correct the Fig. 2 caption.

REPLY: The word "alkalinisation" has been corrected in Figure 2 caption.

Appendix E

Dear editor and reviewer:

Thank you for your valuable advice. In light of the reviewer's comments, we have made the necessary changes to the manuscript, which have been marked in red. Point-by-point responses to the reviewer's comments are provided below.

Response to reviewer's comments

Manuscript ID: RSOS-201904

TITLE: Improved antibacterial activity of hemp fibres by covalent grafting of quaternary ammonium groups

Reviewer: 1

Comments to the Author(s)

The captions of the Figures 2, 3, 4 and 5 are not clear, rewrite them. For example: Figure 2. Effect of NaOH concentration (a) and soaking time (b) on the weight loss and aldehyde group content.

REPLY: The captions of the Figures 2, 3, 4 and 5 have been modified as follows:

Figure 2. Effect of NaOH concentration (a) and soaking time (b) on the weight loss and aldehyde group content of alkalisiation reaction.

Figure 3. Effect of NaIO₄ concentration (a), contact time (b), temperature (c) and pH (d) on the weight loss and aldehyde group content of oxidation reaction.

Figure 4. Effect of solvents (a), glutaraldehyde dosage (b), reaction time (c) and temperature (d) on the weight loss of amination reaction.

Figure 5. Effect of solvents (a), temperature (b), GTA dosage (c) and reaction time (d) on the weight loss of quaternisation reaction.

Fig. 2 and 3: "aldehyde group content", not "aldehyde content". The authors change only the ordinates' captions. Correct the explanation within Figures 2 and 3.

REPLY: We are terribly sorry for our carelessness. And we have revised them in the whole manuscript.

The authors reported the following chemical composition of hemp fibres: (cellulose > 90%, hemicellulose 2-3%, lignin <1%, pectin <1%, lipids <1%). According to that, the total content of non-cellulosic components is <6%. However, the weight loss obtained after the sodium hydroxide treatment is very high (between 15 and 30%). Why?

There is a significant lack of scientific knowledge related to the hemp fibres' chemical composition and the effect of different chemical treatments on hemp fibres. The authors stated: "For shorter and thinner fibre filaments, in addition to the dissolution of pectin and other substances in the fibres, it will also cause the decomposition of the fibre itself, and disperse into the lye. So the weight loss rate of the fibre we measured is higher." ... First of all, hemp fibres are staple fibres, not filament fibres... Second, in the first revision, the authors confirmed that the decomposition of cellulose after the sodium treatment was not occurred. Now, they say that due to the fibre decomposition, the weight loss after the alkali treatment occurs...

REPLY: The fibre used in this study has undergone a series of treatment processes and can be directly used in textiles. The retention of non-cellulose components is less than 10% to improve its performance. These processes include not only chemical treatments such as degumming, but also physical treatments such as mechanical carding, which will have a certain impact on the strength of hemp fibre. Especially in the process of mechanical carding, although most of the short-fibre filaments will be removed, some long-fibre filaments or branched filaments will also be mechanically damaged due to excessive stretching, resulting in a decrease in fibre strength or even breakage. Therefore, there are still some short and thin filaments in the fibre. In our re-chemical treatment process, these short and thin, mechanical damage, low-strength fibre filaments will also shed from the fibre bundle due to the action of strong alkali, dispersed into the lye, at the same time resulting in a high fiber weight loss (<10%, Figure 2). After careful consideration, it is inappropriate to use the term "decomposition" here, so we have chosen the term "shedding" to describe this phenomenon.

There are still exist three different forms: alkalisation, alkalisation and alkalization. Choose one of them and make appropriate corrections in the manuscript.

REPLY: Thanks for your meticulousness! We have corrected it in the manuscript.

P9, L7: Remove one "peaks".

REPLY: The repeated "peaks" has been removed.

P9, L12: hemp molecules?!

REPLY: Sorry for that, it should be "hemp fibre".

In the section characterization, the authors reported on the influence of lignin dissolution on the peak intensity. If you have <1% lignin, it can not affect the peak intensity.

REPLY: We have repeatedly confirmed that the lignin content in hemp fibre used in this study is less than 1%. At the same time, we have studied your comments carefully and we agree with your view that the peak strength will not be affected if the lignin content is less than 1%. Therefore, in order to ensure the scientific and rigorous research analysis, we have deleted the relevant content.

The sentence "If the solution is red, 0.03 mol/L sodium hydroxide methanol solution was added dropwise..." construction is not correct. English should be improved in the whole manuscript.

REPLY: The sentence was modified as "If the solution turned red, the sodium hydroxide methanol solution of 0.03 mol/L was added dropwise to make it yellow". Furthermore, we have revised the whole manuscript carefully again.